# Secondary inorganic aerosols in Europe: sources and the significant influence of biogenic VOC emissions especially on ammonium nitrate

Sebnem Aksoyoglu, Giancarlo Ciarelli, Imad El-Haddad, Urs Baltensperger and André S. H. Prévôt

5    Laboratory of Atmospheric Chemistry, Paul Scherrer Institute, 5232 Villigen PSI, Switzerland
*Correspondence to*: Sebnem Aksoyoglu (sebnem.aksoyoglu@psi.ch)

**Abstract** Contributions of various anthropogenic sources to the secondary inorganic aerosol (SIA) in Europe as well as the role of biogenic emissions on SIA formation were investigated using the three-dimensional regional model CAMx (Comprehensive air quality model with extensions). Simulations were carried out for two periods of EMEP field campaigns (February-March 2009 and June 2006), which are representative of cold and warm seasons, respectively. Biogenic volatile organic compounds (BVOCs) are known mainly as precursors of ozone and secondary organic aerosol (SOA), but their role on inorganic aerosol formation has not attracted much attention so far. In this study, we showed the importance of the chemical reactions of BVOCs and how they affect the oxidant concentrations leading to significant changes especially in the formation of ammonium nitrate. A sensitivity test with doubled BVOC emissions in Europe during the warm season showed a large increase in secondary organic aerosol (SOA) concentrations (by about a factor of two) while particulate inorganic nitrate concentrations decreased by up to 35% leading to a better agreement between the model results and measurements. Sulfate concentrations decreased as well, the change, however, was smaller. The changes in inorganic nitrate and sulfate concentrations occurred at different locations in Europe indicating the importance of precursor gases and biogenic emission types for the negative correlation between BVOCs and SIA. Further analysis of the data suggested that reactions of the additional terpenes with nitrate radicals at night were responsible for the decline in inorganic nitrate formation, whereas oxidation of BVOCs with OH radicals led to a decrease in sulfate. Source apportionment results suggest that the main anthropogenic source of precursors leading to formation of particulate inorganic nitrate is road transport (SNAP7), whereas combustion in energy and transformation industries (SNAP1) was the most important contributor to sulfate particulate mass. Emissions from international shipping were also found to be very important for both nitrate and sulfate formation in Europe. In addition, we examined also contributions from the geographical source regions to SIA concentrations in the most densely populated region of Switzerland, the Swiss Plateau.

## 1 Introduction

Particulate matter (PM) is known to have adverse effects on human health and climate, and is still a problematic pollutant in Europe in spite of considerable improvements in the last decades (Barmpadimos et al., 2012; EEA, 2014). The sources and evolution of PM in the atmosphere are among the most extensively investigated topics in current atmospheric research (Fuzzi et al., 2015; Denier van der Gon et al., 2015). PM is either directly emitted or formed in the atmosphere as secondary inorganic (SIA) and organic aerosols (SOA). The main precursor gases for SIA are $SO_2$,

$NO_x$ and $NH_3$, which react in the atmosphere to form ammonium sulfate and nitrate compounds. Observations from the EMEP network show that SIA concentrations in Europe increase from north to south, with an average contribution of 34% to $PM_{10}$ (particles with an aerodynamic diameter $d <$ 10 µm) mass (Aas et al., 2012). Earlier studies suggest that SIA constitutes more than half of $PM_{2.5}$ ($d < 2.5$ µm) concentrations in Europe, especially in winter, and ammonium nitrate is the dominant component of SIA in western and central Europe (Schaap et al., 2004; Aksoyoglu et al., 2011; 2012; Squizzato et al., 2013). A combination of meteorological conditions and various emission sources led to highly elevated PM concentrations in Europe during early spring episodes in the past, mainly due to high ammonium nitrate concentrations (Sciare et al., 2010; Revuelta et al., 2012). Knowing the location and strength of sources contributing to $PM_{2.5}$ is essential for developing effective control strategies. In spite of the fact that the formation mechanisms of SIA are better understood than those of organic aerosols, chemical transport models (CTMs) still have difficulties to capture measured concentrations. This is usually attributed to uncertainties in $NH_3$ emissions (Aan de Brugh et al., 2011; Wang et al., 2013) while the effect of uncertainties in $NO_x$ emissions and transformation cannot be ruled out (Vaughan et al., 2016). Modeling the formation of the semi-volatile ammonium nitrate is difficult because it is strongly dependent on the ambient conditions. On the other hand, a lot of effort is being made on understanding the formation of SOA and the role of BVOC emissions on organic nitrates, but the indirect effect of BVOC emissions on the formation of inorganic nitrate (ammonium nitrate) has so far not attracted any attention. Biogenic species such as isoprene, mono- and sesquiterpenes emitted from vegetation are known mainly as precursors of secondary pollutants like ozone and SOA (Kanakidou et al., 2005; Sartelet et al., 2012). The nitrate radical is an effective nocturnal oxidizer of VOCs and it is especially reactive towards biogenic volatile organic compounds (BVOCs). Laboratory experiments showed a rapid production of SOA with high yields when some monoterpenes were oxidized by nitrate radicals (Fry et al., 2011; Boyd et al., 2015). Reactions of isoprene lead to the formation of SOA mainly during the daytime while nighttime oxidation of monoterpenes by the nitrate radical is responsible for organic nitrate formation (Ayres et al., 2015; Kiendler-Scharr et al., 2016; Xu et al., 2015a,b). Atmospheric reactions of BVOC species might change the oxidant concentrations significantly, affecting the formation of secondary compounds. In many areas in Europe, models overestimate ammonium nitrate concentrations during nighttime while SOA is underestimated especially during daytime hours (Prank et al., 2016; Knote et al., 2011; Colette et al., 2011; de Meij et al., 2006). Among other issues such as uncertainties in anthropogenic precursor emissions, deposition and missing emission sources, one should also consider the sensitivity of the secondary inorganic aerosol formation to BVOC emissions. Biogenic emissions are generated by emission models (e.g. MEGAN, Guenther

et al., 2012; BEIS (https://www.epa.gov/air-emissions-modeling/biogenic-emission-inventory-system-beis); Simpson et al., 1999; Steinbrecher et al., 2009) to be used in CTMs and the resulting emissions vary significantly depending on the model used or even on the land cover used within the same model (Huang et al., 2015). BVOC emissions are known to have very large uncertainties (Sindelarova et al., 2014; Emmerson et al., 2016) and therefore, their role in the formation of secondary inorganic aerosols might be quite significant.

Although there has been extensive research on the formation of SOA from the oxidation of BVOCs (Carlton et al., 2009; Hallquist et al., 2009; Ayres et al., 2015; Xu et al., 2015a; Fuzzi et al., 2015), to our knowledge, effects of BVOCs on SIA, especially on ammonium nitrate, have been scarcely investigated (Karambelas, 2013). Several studies so far emphasized the significance of BVOC reactions with nitrate radicals as leading to "anthropogenically influenced biogenic SOA" (Ng et al., 2016). In this study we show another consequence –although with smaller influence- of such reactions leading to "biogenic influence on anthropogenic ammonium nitrate" in Europe.

## 2 Modeling Methods

### 2.1 Air quality model CAMx

In this study, we used the regional air quality model CAMx-v5.40 with its PSAT (Particulate Source Apportionment Technology) tool (ENVIRON, 2011). The model domain covered Europe using latitude-longitude geographical coordinates with a horizontal resolution of 0.25° x 0.25°. We used 33 terrain-following σ-levels up to about 350 hPa. The Carbon Bond (CB05) gas phase mechanism (Yarwood et al., 2005) was used and partitioning of inorganic aerosols (sulfate, nitrate, ammonium, sodium and chloride) was performed using the ISORROPIA thermodynamic model (Nenes et al., 1998). Aqueous sulfate and nitrate formation in cloud water was simulated using the RADM aqueous chemistry algorithm (Chang et al., 1987). Partitioning of condensable organic gases to secondary organic aerosols (SOA) was calculated using the semi-volatile equilibrium scheme called SOAP (Strader et al., 1999). SOA precursor species and reactions are given elsewhere (Aksoyoglu et al., 2011). Removal processes including dry and wet deposition were simulated using the Zhang resistance model (Zhang et al., 2003) and a scavenging model approach for both gases and aerosols (ENVIRON, 2011).

Input parameters for CAMx were provided by INERIS within the EURODELTA III project (Bessagnet et al., 2016). Hourly three-dimensional meteorological fields for wind speed and direction, pressure, temperature, specific humidity, cloud cover and rain were calculated from ECMWF IFS (Integrated Forecast System) data at 0.2° resolution within the EURODELTA III

exercise. MACC (Monitoring Atmospheric Composition and Climate) reanalysis data were used to generate initial and boundary condition fields (Benedetti et al., 2009; Inness et al., 2013). Photolysis rates were calculated using the Tropospheric Ultraviolet and Visible (TUV) Radiation Model (https://www2.acom.ucar.edu/modeling/tropospheric-ultraviolet-and-visible-tuv-radiation-model). The ozone column densities to determine the spatial and temporal variation of the photolysis rates were extracted from TOMS data (https://ozoneaq.gsfc.nasa.gov/data/omi/). Anthropogenic emissions were prepared by merging different emission databases such as TNO-MACC (Kuenen et al., 2011), EMEP (Vestreng et al., 2007) and GAINS (http://gains.iiasa.ac.at/gains) as described in Bessagnet et al. (2016). We calculated the gridded biogenic VOC emissions using the Model of Emissions of Gases and Aerosols from Nature (MEGAN v2.1) (Guenther et al., 2012) driven by the meteorological variables.

We ran CAMx with PSAT for the two EMEP intensive measurement campaign periods: 25 February-26 March 2009 (cold season) and 1-30 June 2006 (warm season) with a 14-day spin-up before each period. The model results for aerosols in this study refer to $PM_{2.5}$ fraction. In order to investigate the role of biogenic emissions on the SIA formation, we doubled the BVOC emissions in June 2006 in the model domain and repeated the simulations. We analyzed the model results by means of the Chemical Process Analysis (CPA) tool of CAMx, which provides detailed reaction rate information for selected species from various chemical reactions.

## 2.2 Particulate Source Apportionment Technology (PSAT)

Source apportionment techniques are used to identify the sources of atmospheric pollutants (Viana et al., 2008). It is relatively simple to apportion primary PM among its sources using any pollution model because their source-receptor relationships are linear. On the other hand, Eulerian models are better suited to model secondary PM because they account for chemical interactions among sources. The CAMx tool PSAT was used to assess the contribution of different geographic regions and source categories to modeled concentrations of SIA. PSAT uses reactive tracers to apportion primary and secondary PM as well as the gaseous precursors among different source categories and regions. A single tracer can track primary PM species, whereas secondary PM species require several tracers because of the more complex relationship between gaseous precursors and the resulting particles. PSAT assumes that PM should be apportioned to the primary precursor for each type of particle. Thus particulate sulfate ($PSO_4$) is apportioned to $SO_x$ emissions, particulate nitrate ($PNO_3$) is apportioned to $NO_x$ emissions, and particulate ammonium ($PNH_4$) is apportioned to $NH_3$ emissions (ENVIRON, 2011). One has to keep in mind that PSAT provides a PM attribution to

source regions and categories for a given emissions matrix, but does not provide quantitative information as to how PM contributions would change as emissions are altered because chemical interactions are nonlinear. We defined source categories (Table 1) based on the SNAP codes given by Kuenen et al. (2011). We also modeled the contribution from the boundaries of the domain as a non-European source.

In addition to source categories, we also investigated in a case study the contribution from various source regions to SIA concentrations in the Swiss Plateau, which is the most densely populated part of Switzerland, comprising the area between the Jura Mountains and the Swiss Alps (Fig. S1). The selected source regions were identified as Switzerland (domestic), France, Germany, Italy and Austria (surrounding countries), Poland and Benelux countries (polluted regions), Sea (marine areas), Rest (rest of the domain), and BC (boundary conditions) (see Fig. S2).

Measurements with the high resolution AMS (aerosol mass spectrometer) are available at 11 European sites for the February-March 2009 period (Crippa et al., 2014). A detailed evaluation of CAMx model performance for the two periods simulated in this study can be found elsewhere (Ciarelli et al., 2016). Since AMS data are only available at Payerne for the summer June 2006 period, we evaluated meteorological parameters and SIA concentrations at that site in both cold and warm seasons.

## 3 Results and discussion

### 3.1 Model evaluation

Model performance was evaluated using methods implemented in the Atmospheric Model Evaluation Tool (AMET; Appel et al., 2011). The modeled concentrations of the gas phase species and $PM_{2.5}$ were compared with the measurements from the AirBase database (see soccer-goal plots in Fig. 1). Only rural-background stations with at least 80% daily average observations available were considered for the evaluation. The model performance for the gaseous species in June 2006 was similar to another study performed during the AQMEII exercise in July 2006 using the CAMx model (Nopmongcol et al., 2011). Although $NO_2$ was underestimated in both cases, fractional bias (FB) and fractional error (FE) were smaller in our study (AQMEII: FB -62, FE 73; this study; FB -42, FE 60). The positive bias for $SO_2$ in both studies is likely due to vertical distribution of emissions as reported by Nopmongcol et al. (2011). The performance for $NO_2$ in February-March 2009 was better than in summer. The tendency to underestimate $NO_2$ in summer was reported in other modeling studies as well (Bessagnet et al., 2016, Knote et al., 2011) and may result from either insufficient NOx emissions or from too high measurements (very close to sources or by

artefacts from other oxidized nitrogen compounds in the $NO_2$ measurements) (Steinbacher et al., 2007; Vaughan et al., 2016). Ozone concentrations were captured well in the cold season while there was a slight overestimation in June 2006.

The model performance for $PM_{2.5}$ was very good; recommended model performance criteria (MFB ≤ ± 60%, MFE: ≤ +75%) as well as the performance goal (MFB ≤ ± 30%, MFE ≤ +50%) by Boylan and Russell (2006) were achieved in both periods (see Fig. 1). The components of $PM_{2.5}$ were further evaluated for the period in February-March 2009 where high-resolution AMS measurements at 11 European sites were available (Table 2). In general, there is a tendency to overestimate the inorganic aerosols and underestimate the organic aerosols. The detailed discussion for the evaluation of nitrate, sulfate and ammonium at all sites can be found in Ciarelli et al. (2016) where the input parameters and the model were the same except the VBS (Volatility Basis Set) approach used to model organic aerosols.

The components of $PM_{2.5}$ were also evaluated for the period in June 2006 at Payerne (Switzerland) where the AMS measurements were available. Since Payerne is a representative site for the Swiss Plateau, evaluation of model performance at that location is also important for the case study discussed in Section 3.6. The temporal variations of both meteorological parameters and chemical components were captured quite well by the model in both periods (Figs. S3-S4). There is a clear correlation between the SIA concentrations and wind speed. In the cold season, agreement between modeled and measured wind speed is very good (Fig. S3); concentrations are higher when the wind speed is low (25 February-5 March, 18-19 March, and 22-23 March). The modeled SIA concentrations are very close to the measured ones, except for a few days (3 and 23 March). As part of the EURODELTA III project, Bessagnet et al. (2016) analyzed measured and modeled meteorological variables such as PBL height and wind speed at several sites in Paris. The study suggested that observations were well reproduced by ECMWF IFS in general except for a few days when the PBL height was overestimated (Bessagnet et al., 2016). The modeled SIA concentrations during the warm season were also very close to measurements except for an overestimation during the first week when temperatures were relatively lower (Fig. S4, right panel). Underestimated wind speed and PBL height might be some of the reasons of this discrepancy. A slight underestimation of temperature might also have caused more partitioning on the particle phase.

**3.2 Particulate Nitrate ($PNO_3$)**

The modeled $PNO_3$ concentrations were higher in the cold season with a monthly average of up to 14 µg m$^{-3}$ over northern Italy (Fig. 2, top left). In the warm season, highest concentrations (7-8 µg

m$^{-3}$) were predicted mainly around the English Channel and Benelux area (Fig. 2, top right). The largest contribution to nitrate was from road transport, followed by ships and combustion in energy and transformation industries while the contribution from the boundaries was very small (Fig. 2). The relative contribution from road traffic was higher in Eastern Europe (Fig. S5). The contribution of SNAP 1 sources to PNO$_3$ was higher in East European countries during the cold season (Fig. 2). The relative contribution from offshore petroleum activities in the North Sea to PNO$_3$ was quite high as seen in Fig. S5. Emissions from the petroleum sector are generally exhaust gases from combustion of natural gas in turbines, flaring of natural gas and combustion of diesel. Ship emissions (SNAP8) in the warm season led to PNO$_3$ formation mainly along the English Channel, whereas their contribution in the cold season was predicted throughout the whole of Central Europe, most likely because of higher NH$_3$ emissions in this area in the early spring. This agrees with the results of previous simulations (Aksoyoglu et al., 2016). On the other hand, the relative contribution from ship emissions to PNO$_3$ was clearly higher over the Mediterranean in both seasons (Fig. S5).

### 3.3 Particulate Sulfate (PSO$_4$)

The modeled particulate sulfate concentrations were relatively low over central Europe during the cold season (Fig. 3). A significant contribution, however, was detected from the eastern boundaries of the model domain (for relative contributions see Fig. S6). The effect of boundary inflows on sulfate levels in the warm season was much lower. MACC reanalysis data have already been evaluated in detail (e.g. Inness et al., 2013; Giordano et al., 2015). Evaluations during the AQMEII-2 exercise showed a positive bias for sulfate and suggested that it was because MACC aerosol model does not contain a representation of ammonium nitrate aerosol which represents a large component of the European aerosol loading (Giordano et al., 2015). Therefore the assimilation of satellite AOD will tend to increase the other aerosol components to give the correct AOD overall. In our study, high sulfate at the eastern boundary was mainly during the February-March 2009 period and affected only the eastern part of the domain while the rest of Europe was not affected.

Emissions from the combustion in energy and transformation industries are the main sources for particulate sulfate in Eastern Europe, while shipping contributes mainly in the Mediterranean and along other shipping routes around Europe (Fig. 3). Significant contributions from SNAP1 sources in June were predicted over the Balkan countries as well as in northwest Spain (Fig. S6) where large facilities are grouped (Guevara et al., 2014). The contribution from ships to PSO$_4$ was predicted to be larger during the warm season. Although ship emissions are only slightly higher in summer, their larger contribution to sulfate is mainly due to higher oxidation potential in the warm season (Aksoyoglu et al., 2016).

### 3.4 Particulate Ammonium (PNH$_4$)

The modeled PNH$_4$ concentrations were relatively higher during the period of February-March 2009 since its main source is agriculture with largest emissions occurring in early spring (Fig. 4, see Fig. S7 for relative contributions). The highest PNH$_4$ concentrations were predicted in central Europe. A small contribution (2-10%) from road transport could be seen around the urban areas (Fig. 4, Fig.S7).

### 3.5 Role of biogenic VOC emissions

Biogenic VOC emissions are known as effective SOA precursors. There are large uncertainties associated with BVOC emission estimates due to the substantial number of compounds and biological sources (Guenther, 2013; Oderbolz et al., 2013). The gas-phase reactions of biogenic species used in the chemical mechanism CB05 in CAMx are given in Table S1. Some of the
oxidation reactions produce condensable gases (CG) that might lead to formation of SOA particles (Table S2). The monthly average emissions of isoprene, monoterpenes and sesquiterpenes and their contributions to SOA concentrations in the warm season are shown in Fig. S8. Although isoprene emissions were larger (especially in southern Europe), more SOA was produced by sesquiterpenes and monoterpenes due to their higher SOA yields (Lee et al., 2006; Hallquist et al., 2009;
ENVIRON, 2011). SOA was predominantly produced by biogenic precursors and oligomerization processes; the contribution of anthropogenic precursors to SOA was very small.

The model results showed a large increase in SOA (Fig. S9) when BVOC emissions were doubled while PNO$_3$ and PSO$_4$ concentrations decreased (Fig. 5). Since the positive correlation between biogenic emissions and SOA is relatively well known, we focused on the role of BVOCs on SIA
formation. Increased BVOC emissions led to greater decreases in PNO$_3$ (up to 35%, Fig. S10, left panel) than in PSO$_4$ (<12%, Fig. S10, right panel). The largest decrease in PNO$_3$ occurred around the Benelux area and northern Italy where concentrations were highest (Fig. 5). PSO$_4$ decreased mostly in Eastern Europe where SO$_2$ concentrations were relatively higher. In a study in the eastern U.S, Karambelas (2013) also reported a negative correlation between BVOC emissions and SIA.
From the comparison of the base case and no-biogenic emission simulations, the author attributed the increase in SIA concentrations to the increased availability of OH radicals (as a result of elimination of SOA in the absence of BVOC emissions) for oxidation of precursor gases such as NO$_2$ and SO$_2$.

We analyzed our results in June 2006 further by investigating the changes in OH radical and
production rate of organic nitrates and nitric acid (HNO$_3$) from the two main reaction pathways:

NO$_2$ + OH (daytime) and N$_2$O$_5$ + H$_2$O (nighttime) in one grid cell (Payerne, Switzerland) for three different cases: 1) with standard BVOC emissions, 2) with doubled BVOC emissions, 3) with doubled BVOC emissions and without BVOC+NO$_3$ reactions (see Table S1 for the reactions of BVOCs used in CAMx). As seen in Figs. 6a and 6b, the daytime production of HNO$_3$ is higher than the nighttime production. On the other hand, the deposition rate of HNO$_3$ is much higher during the day than in nighttime (Fischer et al., 2006; Phillips et al., 2006; Zhou et al., 2010). HNO$_3$ concentrations usually show a distinct minimum during early morning hours and a maximum in the afternoon with a rapid drop near sunset (Fischer et al., 2006; Aas et al., 2012). The diurnal cycle of the modeled HNO$_3$ in this study shows a similar behavior (Fig. S11). Switching off the nighttime hydrolysis of N$_2$O$_5$ in an additional model simulation led to a significant decrease in PNO$_3$ concentrations at night (Fig. 7, upper panel). The overall effect of hydrolysis reaction in central Europe during June 2006 was predicted to vary between 20-46% (0.5 and 2.8 µg m$^{-3}$) as seen in the lower panel of Fig. 7 (see Fig. S12 for the absolute changes). These results suggest that an important part of HNO$_3$ leading to the formation of ammonium nitrate is produced from the hydrolysis of N$_2$O$_5$ at night. The low temperatures at night then keep the nitrate in the particle phase without evaporating back to HNO$_3$.

OH radical consumption increased when BVOC emissions were doubled, mainly during the daytime (Fig. S13) due to OH oxidation reactions of BVOCs. The decrease in OH radicals most likely caused a reduction in SO$_2$ oxidation leading to a decrease in PSO$_4$ concentrations (see Fig. 5, right panel). On the other hand, HNO$_3$ production from daytime (Fig. 6a) and nighttime (Fig. 6b) reactions as well as PNO$_3$ concentrations (Fig. 6d) decreased with increasing BVOC emissions. The decrease in HNO$_3$ production via daytime reaction indicates a decrease in available OH radicals due to BVOC+OH reactions (Table S1 and Fig. S13). Switching off the reactions with NO$_3$ radical did not affect the daytime production further (Fig. 6a) as expected since NO$_3$ is a nocturnal oxidant (e.g. Platt et al., 1981; Russell et al., 1986; Platt and Heintz, 1994; Seinfeld and Pandis, 2012). On the other hand, HNO$_3$ production at night decreased with increased BVOC emissions suggesting that the available NO$_3$ radicals decreased due to BVOC + NO$_3$ reactions (Fig. 6b). The fact that HNO$_3$ formation at night increased significantly when BVOC + NO$_3$ reactions were switched off presents further evidence that BVOC+NO$_3$ reactions were responsible for the changes in PNO$_3$ concentrations (Fig. 6b, d). Organic nitrates may serve as either a NOx reservoir or a NOx sink (Kiendler-Scharr et al., 2016). Their production increased with doubled BVOC emissions (Fig. 6c) because of isoprene (ISOP) and isoprene oxidation product (ISPD) reactions with NO$_2$ and NO$_3$ and terpene (TERP) reactions with NO$_3$ (see reactions in Table S1). On the other hand, when BVOC

reactions with $NO_3$ radical were switched off, organic nitrate production decreased significantly,

especially at night due to reduced production from terpene $+NO_3$ reactions. Steinbacher et al (2005) showed that isoprene emissions vanish after sunset and isoprene mixing ratios decline quickly due to chemical reactions with $NO_3$, $O_3$ and OH, leaving no isoprene for further reactions during the night. In addition, isoprene oxidation with $NO_3$ radical produces not only organic nitrates (NTR) but $HNO_3$ as well (see Table S1). It is therefore more likely that oxidation of terpenes with $NO_3$ is

the main driver for the BVOC effect on $PNO_3$. Comparison of modeled and measured diurnal variation of $PNO_3$ concentrations shows that increasing BVOC emissions brought modeled results closer to the observations especially at night (Fig. 8). The results of these sensitivity tests suggest that the overestimation of particulate inorganic nitrate might be partly due to too low biogenic VOC emissions.

**3.6 A case study: The Swiss Plateau**

**3.6.1 Cold Season (February-March 2009)**

The modeled SIA concentrations were relatively high during the first few days in March and during the second half of the period (Fig. 9). Low winds from southwest at the beginning of March (Fig. S3, left panel) led to relatively larger contribution from sources in France as well as from domestic

sources to SIA concentrations in the Swiss Plateau. Then, when the wind direction shifted towards the northeast between 16 and 22 March (see Fig. S3), the contribution from Germany became larger.

Domestic sources contribute to $PNO_3$ concentrations (21%) in the Swiss Plateau as much as sources in Germany (18%) and France (24%) during the whole period of February-March 2009 (Fig. 9, top-

320 middle panel). A large fraction of $PNO_3$ originates from road transport (40%), while 22% is from shipping emissions in the coastal areas (Fig. 9, top-right panel). Sources from non-industrial combustion (SNAP2) and combustion in energy and transformation industries (SNAP1) contribute 16% and 12%, respectively.

Boundary conditions are predicted to have the largest contribution to $PSO_4$ in the Swiss Plateau

(24%) followed by sources in France (17%), Germany (13%) and the sea areas (13%) (Fig. 9, middle panels). The Swiss sources contribute only to 11%. The main source categories are combustion in energy and transformation industries (SNAP1) and non-industrial combustion (SNAP2) contributing 33% and 23% of total sulfate, respectively.

In the case of PNH$_4$, domestic sources are clearly the main contributor (73%) followed by the two neighboring countries France (11%) and Germany (10%). These mainly originate from agricultural activities with a small contribution (3%) from road transport (Fig. 9, lower panels).

**3.6.2 Warm Season (June 2006)**

The magnitude of PNO$_3$ and PNH$_4$ concentrations shown in Fig. 10 reflects the temporal evolution of air temperature in June 2006 (Fig. S3, right panel). During the first half of the month, maximum daily temperatures increased from 10$^o$C to about 30$^o$C, while both PNO$_3$ and PNH$_4$ concentrations decreased. Then both temperatures and nitrate concentrations remained almost the same until the end of June. The wind was blowing from northeast and northwest during the first half of June leading to relatively high contributions to PNO$_3$ concentrations in the Swiss Plateau from Germany, France and the Benelux countries. Over the whole period, the largest contribution to nitrate was predicted to be from Germany (30%) followed by Switzerland (18%) and France (14%) (Fig. 10). A significant amount was also predicted from marine areas (13%) and the Benelux countries (10%). Our results also suggest that nearly half of PNO$_3$ originated from road transport (47%), while ship emissions also contributed significantly (about 22%). On the other hand, Switzerland itself was predicted to be the main source for PNH$_4$ concentrations (71%) with some contribution from Germany (13%) in the first half of June due to northerly winds. Almost all of PNH$_4$ (96%) originated from agricultural activities.

Time series and pie charts for PSO$_4$ in Fig. 10 indicate a large contribution to sulfate concentrations from remote areas – boundary conditions (34%), marine areas (16%) and the rest of the domain (17%) – especially in the second half of the month, which experienced various wind directions. The contribution of domestic sources to PSO$_4$ concentrations was very small (3%); the largest contribution among the emission sources was from SNAP1 (45%), followed by SNAP8 (22%).

**4 Conclusions**

In this study, sources of secondary inorganic aerosols in Europe and the role of biogenic emissions on their formation were investigated. Model simulations with CAMx including its PSAT tool were used to estimate the contributions from 10 anthropogenic emission sources as well as from boundary conditions to the concentrations of particulate inorganic nitrate, sulfate and ammonium in Europe during two periods of EMEP measurement campaigns; February-March 2009 (cold season) and June 2006 (warm season).

Road transport (SNAP7) was predicted to be the most important source for $PNO_3$ with the largest contribution during the cold season over northern Italy. Other important sources were ship emissions (SNAP8), which contributed to particulate inorganic nitrate along the English Channel and Benelux area, and combustion in energy and transformation industries (SNAP1) in Central and Eastern Europe. The model results suggested that $PSO_4$ in Europe originated from SNAP 1 sources especially in Eastern Europe. The contribution of ship emissions in the Mediterranean and along busy shipping routes was larger during the warm season. A large contribution to sulfate in the eastern part of the domain during the cold season was attributed to inflow from the boundary. Agricultural activities were the dominant source for $PNH_4$, with a small (2-10%) contribution from road transport.

A case study with the Swiss Plateau as receptor, showed how wind speed and direction affected the contribution from various source areas to the particulate nitrate, sulfate and ammonium concentrations in the area. These results suggested that the contribution from the domestic sources to $PNO_3$ concentrations in the Swiss Plateau was similar to those from Germany and France during the cold season and almost half of it was from road transport. The sources of $PSO_4$ were mostly of foreign origin from combustion in energy and transformation industries with the domestic contributions of 11% and 3% in winter and summer, respectively. The local agricultural activities were the main source of $PNH_4$. One has to keep in mind that these results refer to the emissions matrix used in this study and they might be different if emissions are modified because chemical interactions are nonlinear.

An important outcome of this study was the significant role of biogenic VOC emissions on the SIA formation, especially on particulate inorganic nitrate. The sensitivity tests carried out during the warm season showed a negative correlation between BVOC emissions and SIA concentrations. Increasing BVOC emissions by a factor of two led to a decrease by 35% and 12% in $PNO_3$ and $PSO_4$, respectively. Overestimation of particulate nitrate at night was reduced as a result of increased BVOC emissions, leading to a better agreement with observations. Further investigations using the Chemical Process Analysis tool of CAMx suggested that reactions of terpenes with nitrate radical at night led to a decrease in $PNO_3$ formation (by reducing $HNO_3$) when BVOC emissions were doubled. Although OH radical concentration was reduced by oxidation reactions of BVOCs, decreased daytime oxidation of $NO_2$ with OH did not affect $PNO_3$ concentrations. On the other hand, reduced availability of OH radical for gas-phase oxidation of $SO_2$ caused a decrease in $PSO_4$ concentrations especially over the Aegean and Mediterranean Sea. These results indicated the importance of BVOC emissions not only for secondary organic aerosol formation but also for

inorganic aerosols. Considering the challenges in BVOC emission estimates in addition to uncertainties in $NH_3$ and $NO_x$ emissions, modeled particulate inorganic nitrate concentrations might have larger uncertainties than assumed so far.

## Acknowledgements

We acknowledge INERIS for providing various model input data within the EURODELTA III exercise. Anthropogenic emissions were based on TNO and EMEP inventories and data from the GAINS database of IIASA. We thank the European Centre for Medium-Range Weather Forecasts (ECMWF) for the access to the meteorological and the global air quality model data. Calculations of land use data were performed at the Swiss National Supercomputing Centre (CSCS). We are grateful to Greg Yarwood at RAMBOLL ENVIRON for his valuable support. This work was financially supported by the Swiss Federal Office of Environment (FOEN).

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

Table 1: Source categories used in this study

| Source categories | |
| --- | --- |
| SNAP1 | Combustion in energy and transformation industries |
| SNAP2 | Non-industrial combustion |
| SNAP3 | Combustion in manufacturing industry |
| SNAP4 | Production processes |
| SNAP5 | Extraction and distribution of fossil fuels and geothermal energy |
| SNAP6 | Solvent and other product use |
| SNAP7 | Road transport |
| SNAP8 | Other mobile sources and machinery |
| SNAP9 | Waste treatment and disposal |
| SNAP10 | Agriculture |
| Boundary conditions (concentrations on the lateral model boundaries) | |

Table 2: Statistical analysis of PNO$_3$, PNH$_4$, PSO$_4$ and OA for February-March 2009 at different AMS sites. (MB: mean bias, ME: mean error, MFB: mean fractional bias, MFE: mean fractional error)

| Site | observed (µg m$^{-3}$) | modeled (µg m$^{-3}$) | MB (µg m$^{-3}$) | ME (µg m$^{-3}$) | MFB (-) | MFE (-) |
|---|---|---|---|---|---|---|
| | | | PNO$_3$ | | | |
| Barcelona | 3.6 | 6.0 | 2.4 | 4.1 | 0.43 | 0.95 |
| Cabauw | 2.2 | 6.5 | 4.3 | 4.4 | 0.85 | 0.98 |
| Chilbolton | 2.7 | 3.9 | 1.2 | 2.1 | 0.01 | 0.75 |
| Helsinki | 1.0 | 2.3 | 1.3 | 1.6 | 0.48 | 0.94 |
| Hyytiälä | 0.2 | 1.3 | 1.1 | 1.2 | 0.49 | 1.11 |
| Mace Head | 0.6 | 1.7 | 1.1 | 1.1 | 0.26 | 0.70 |
| Melpitz | 3.1 | 4.9 | 1.8 | 2.7 | 0.47 | 0.72 |
| Montseny | 3.1 | 6.2 | 3.1 | 4.5 | 0.47 | 1.00 |
| Payerne | 3.9 | 6.3 | 2.4 | 3.2 | 0.47 | 0.66 |
| Puy de Dôme | 0.9 | 2.8 | 1.9 | 2.3 | 1.19 | 1.31 |
| Vavihill | 2.8 | 4.3 | 1.5 | 2.5 | 0.27 | 0.79 |
| | | | PNH$_4$ | | | |
| Barcelona | 1.6 | 2.6 | 1.0 | 1.4 | 0.48 | 0.70 |
| Cabauw | 1.0 | 2.6 | 1.6 | 1.6 | 0.92 | 0.94 |
| Chilbolton | 1.3 | 1.9 | 0.5 | 0.9 | 0.34 | 0.59 |
| Helsinki | 0.8 | 1.5 | 0.8 | 0.8 | 0.66 | 0.73 |
| Hyytiälä | 0.4 | 1.0 | 0.6 | 0.7 | 0.61 | 0.77 |
| Melpitz | 1.4 | 2.4 | 1.0 | 1.3 | 0.52 | 0.72 |
| Montseny | 1.7 | 2.7 | 1.0 | 1.6 | 0.45 | 0.75 |
| Payerne | 1.7 | 2.6 | 0.9 | 1.3 | 0.42 | 0.62 |
| Puy de Dôme | 0.7 | 1.2 | 0.6 | 0.9 | 0.88 | 1.11 |
| Vavihill | 1.6 | 2.2 | 0.7 | 1.0 | 0.29 | 0.57 |
| | | | PSO$_4$ | | | |
| Barcelona | 2.7 | 2.3 | -0.4 | 1.2 | -0.15 | 0.44 |
| Cabauw | 1.0 | 1.9 | 0.9 | 1.2 | 0.65 | 0.78 |
| Chilbolton | 1.3 | 2.0 | 0.6 | 1.1 | 0.36 | 0.68 |
| Helsinki | 2.4 | 2.8 | 0.4 | 0.9 | 0.17 | 0.41 |
| Hyytiälä | 1.4 | 2.2 | 0.7 | 1.0 | 0.19 | 0.71 |
| Mace Head | 0.4 | 1.2 | 0.9 | 0.9 | 1.10 | 1.17 |
| Melpitz | 1.1 | 2.5 | 1.4 | 1.7 | 0.53 | 0.82 |
| Montseny | 1.4 | 2.3 | 1.0 | 1.2 | 0.57 | 0.68 |
| Payerne | 1.1 | 2.0 | 1.0 | 1.2 | 0.57 | 0.77 |
| Puy de Dôme | 0.4 | 1.2 | 0.8 | 0.9 | 1.13 | 1.23 |
| Vavihill | 1.6 | 2.6 | 1.0 | 1.2 | 0.27 | 0.57 |
| | | | OA | | | |
| Barcelona | 8.2 | 2.0 | -6.3 | 6.3 | -1.08 | 1.1 |
| Cabauw | 1.2 | 1.0 | -0.3 | 0.5 | -0.18 | 0.49 |
| Chilbolton | 2.4 | 0.6 | -1.8 | 1.8 | -1.14 | 1.15 |
| Helsinki | 2.7 | 2.0 | -0.7 | 1.5 | -0.21 | 0.64 |
| Hyytiälä | 1.3 | 0.7 | -0.7 | 0.7 | -0.69 | 0.72 |
| Mace Head | 0.8 | 0.2 | -0.6 | 0.6 | -0.71 | 0.90 |
| Melpitz | 1.5 | 0.5 | -1.0 | 1.0 | -0.86 | 0.88 |
| Montseny | 3.1 | 2.5 | -0.5 | 1.7 | -0.05 | 0.62 |
| Payerne | 4.1 | 1.1 | -3.0 | 3.0 | -1.03 | 1.07 |

| | | | | | | |
|---|---|---|---|---|---|---|
| Puy de Dôme | 0.6 | 1.0 | 0.4 | 0.7 | 0.56 | 0.92 |
| Vavihill | 3.9 | 1.1 | -2.8 | 2.8 | -1.06 | 1.07 |

665

**Fig. 1: Soccer-goal plots for hourly concentrations of PM$_{2.5}$, NO$_2$, SO$_2$, CO and O$_3$ in June 2006 (left) and February-March 2009 (right). The number of measurement stations (AirBase) is indicated in the legend for each species.**

670

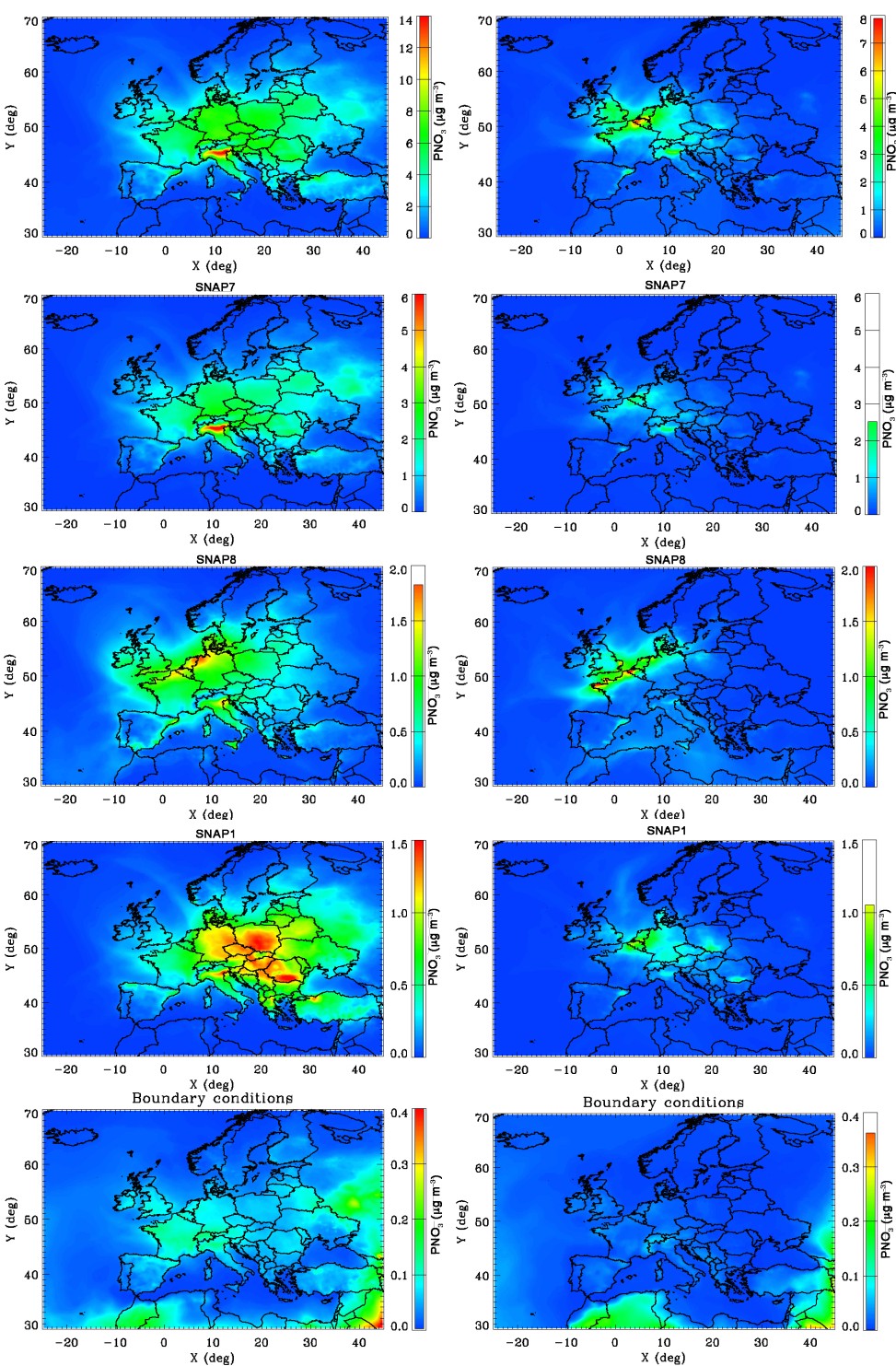


**Figure 2: Average concentrations of PNO₃ and contributions from road transport (SNAP7), ships (SNAP8), combustion in energy and transformation industries (SNAP1) and boundary conditions in February-March 2009 (left) and in June 2006 (right).**


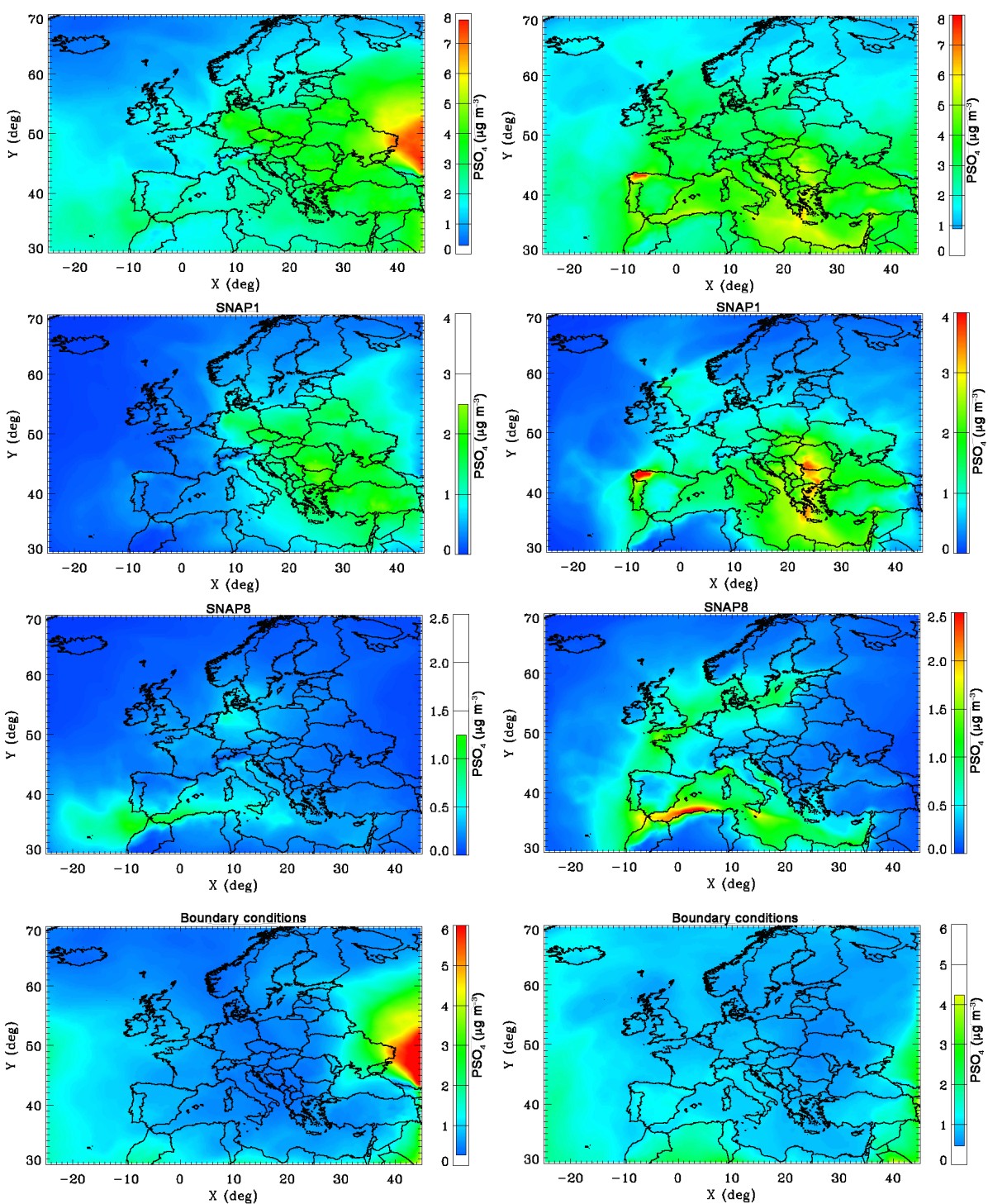

**Figure 3: Average concentrations of PSO₄ and contributions from combustion in energy and transformation industries (SNAP1), ships (SNAP8) and boundary conditions in February-March 2009 (left) and in June 2006 (right).**

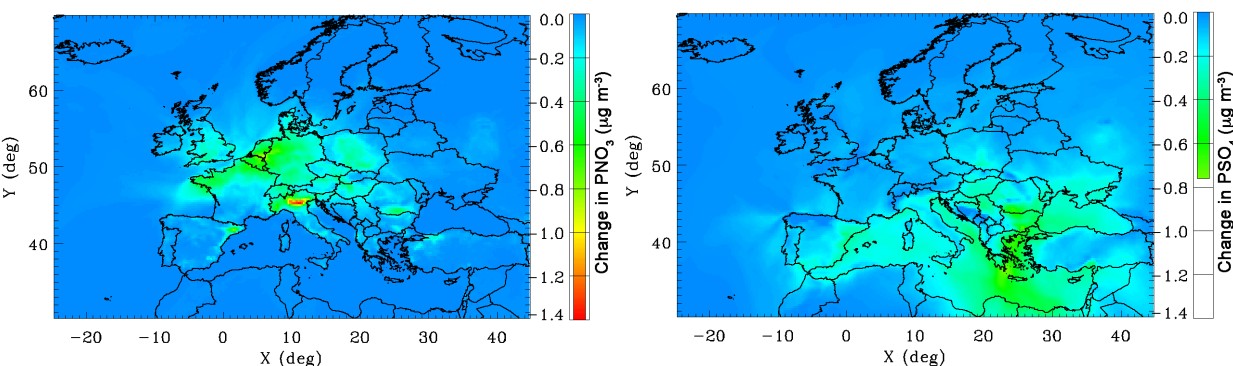


**Figure 4: Average concentrations of PNH₄ and contributions from agriculture (SNAP10) and road transport (SNAP7) in February-March 2009 (left) and in June 2006 (right).**

**Figure 5: Change in PNO₃ (left) and PSO₄ (right) concentrations in June 2006 when BVOC emissions were doubled.**

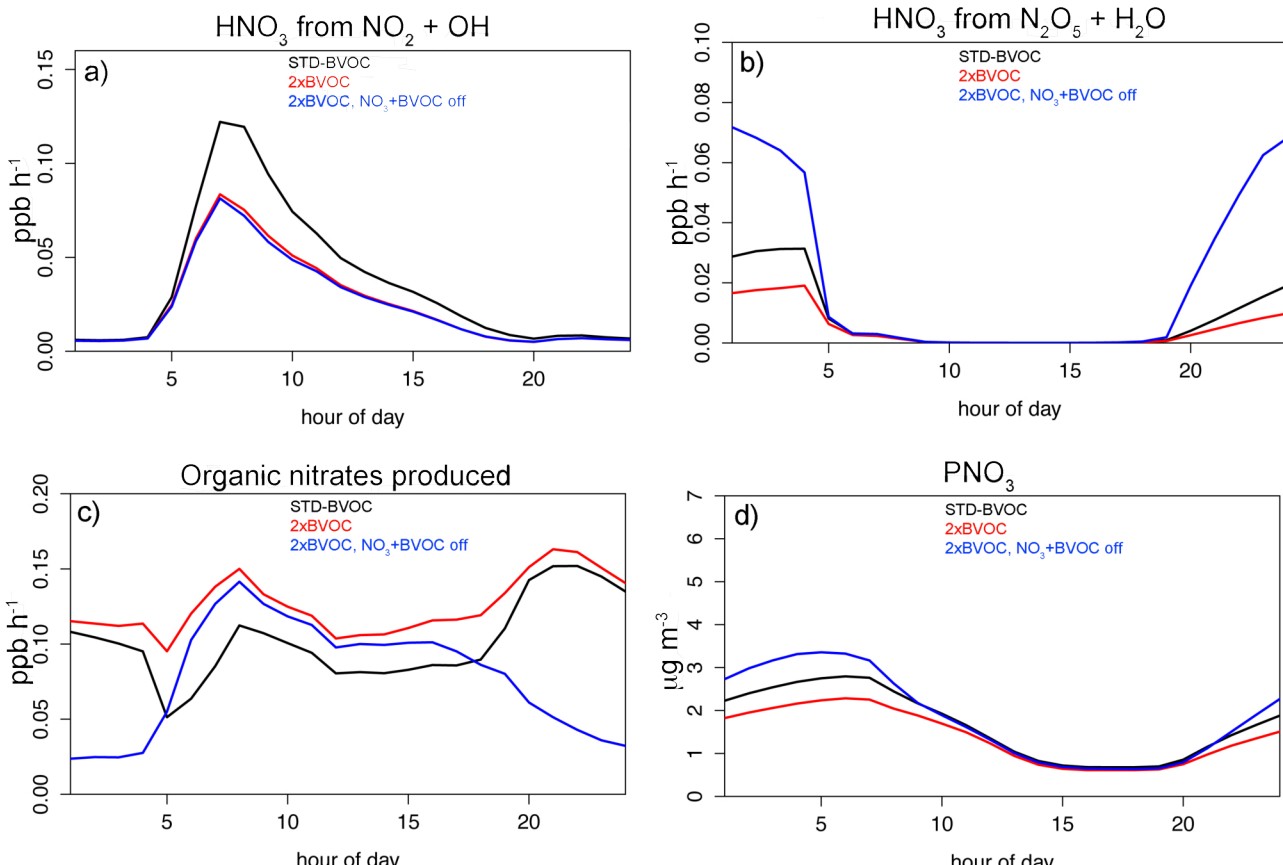

**Figure 6: Changes in diurnal cycle of a) production rate of HNO₃ from daytime reaction, b) production rate of HNO₃ from nighttime reaction, c) production rate of organic nitrates d) particulate nitrate concentrations. Black: with standard BVOC emissions, red: with doubled BVOC emissions, blue: with doubled BVOC emissions and without BVOC+NO₃ reactions (Payerne, average June 2006).**


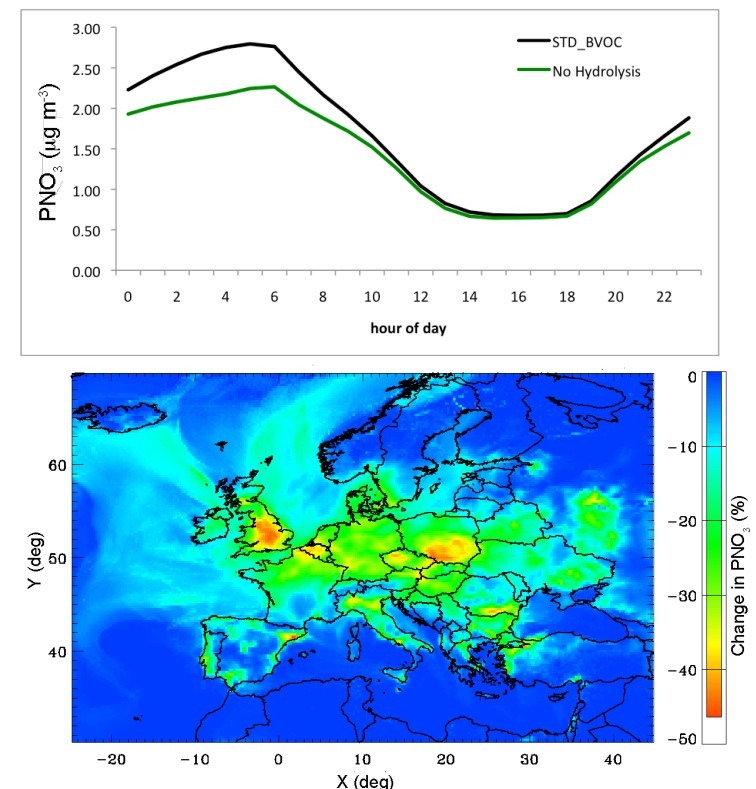

**Figure 7: Upper panel: Change in the diurnal cycle of particulate nitrate concentrations when hydrolysis reaction of N$_2$O$_5$ was switched off (Payerne, average June 2006) Lower panel: Change in PNO$_3$ concentrations (in %) in whole model domain when hydrolysis reaction of N$_2$O$_5$ was switched off (June 2006).**

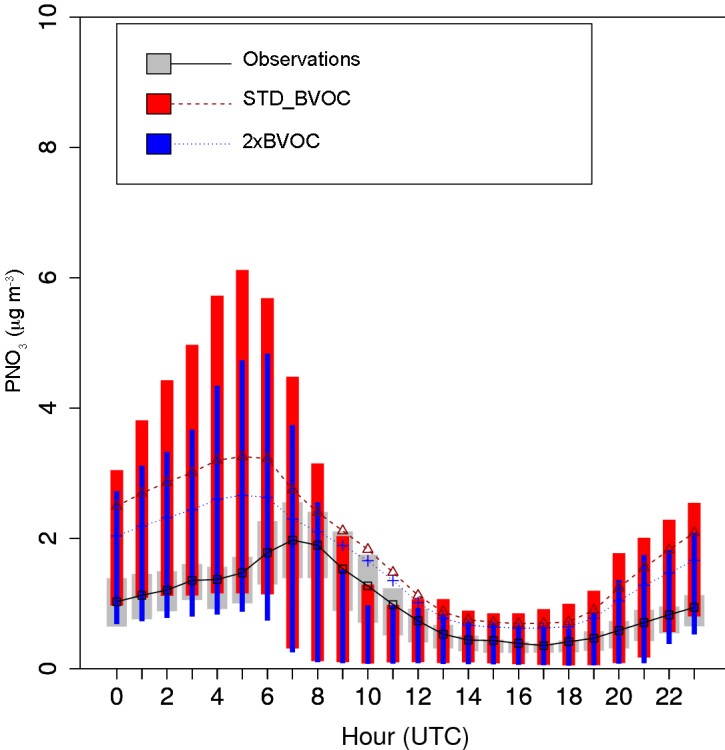

**Figure 8: Hourly box plots showing the diurnal cycle of observed (grey) and modelled particulate nitrate concentrations with standard BVOC emissions (red) and with doubled BVOC emissions (blue) in June 2006 at Payerne. Bars show the 25$^{th}$ and 75$^{th}$ quantiles while the mean is displayed by the line.**

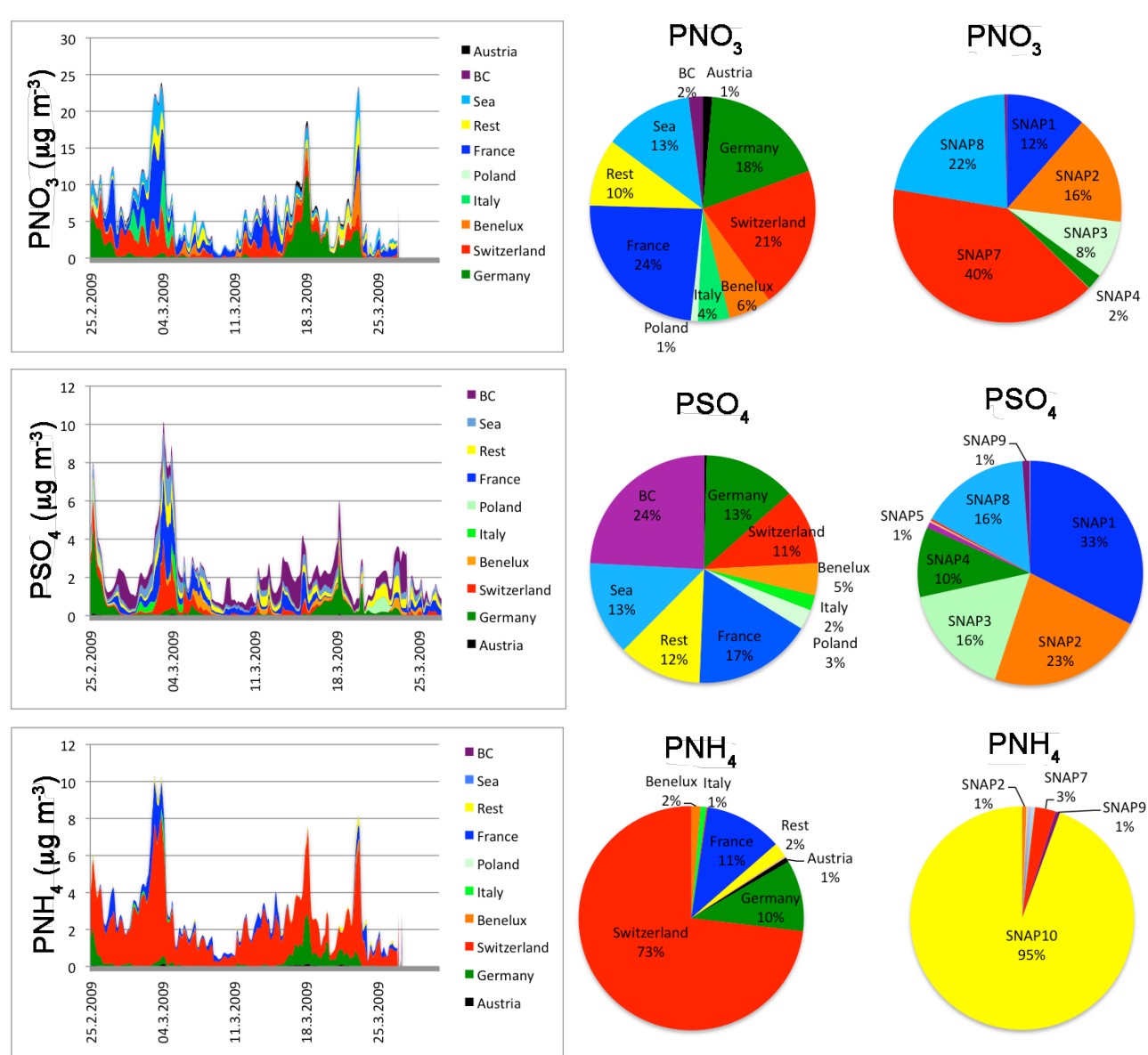

**Figure 9: Contributions from various source regions (time series on the left, pie-charts in the middle) and categories (pie-charts on the right) to the concentrations of PNO$_3$ (top), PSO$_4$ (middle) and PNH$_4$ (bottom) in the Swiss Plateau during February-March 2009.**


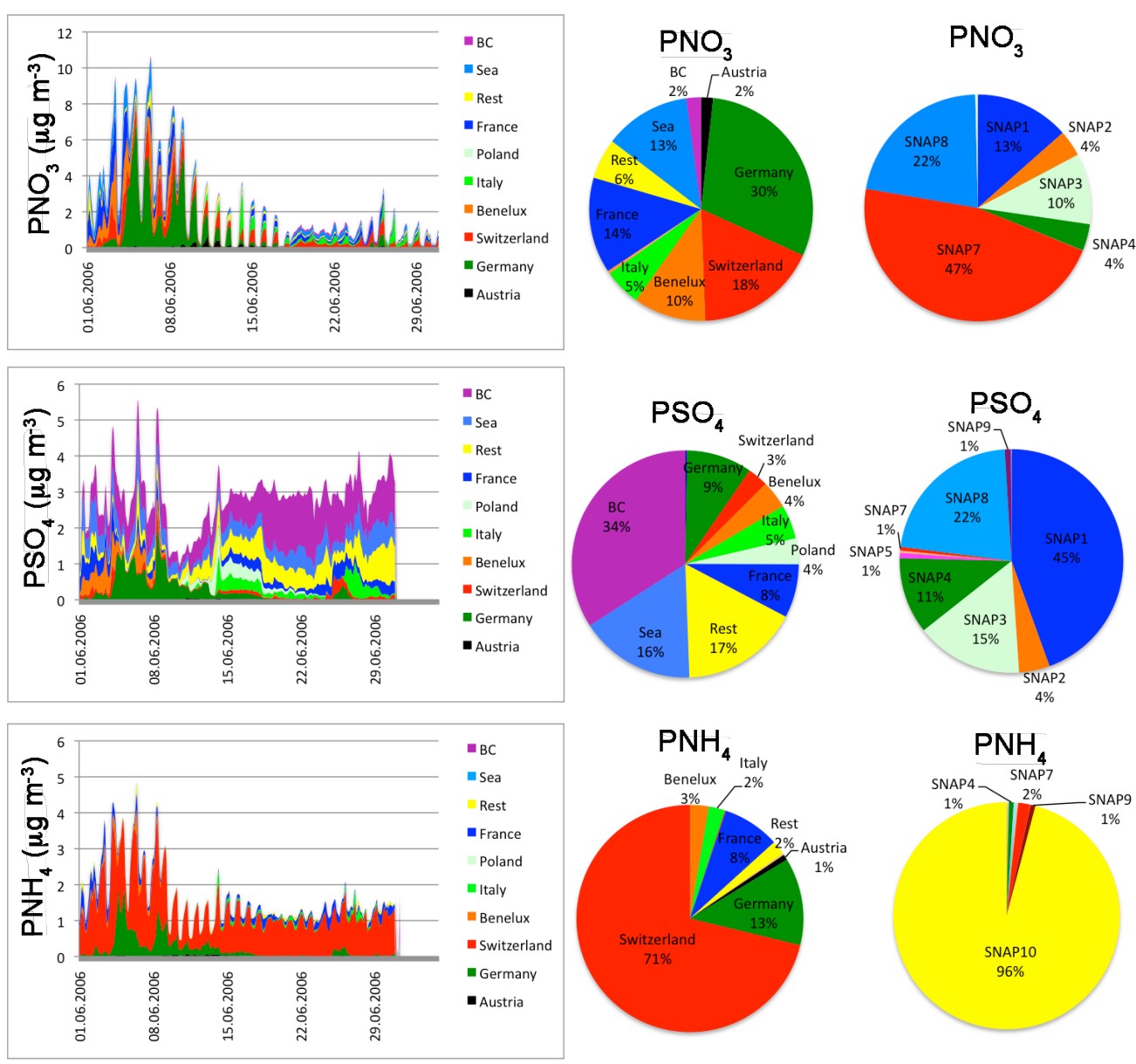

**Figure 10: Contributions from various source regions (time series on the left, pie-charts in the middle) and categories (pie-charts on the right) to concentrations of PNO$_3$ (top), PSO$_4$ (middle) and PNH$_4$ (bottom) in the Swiss Plateau during June 2006.**
