# Peer review of "Secondary inorganic aerosols in Europe: sources and the significant influence of biogenic VOC emissions especially on ammonium nitrate"

_Atmospheric Chemistry and Physics, 2016_

## Referee Comment (RC1) · Anonymous Referee #1 · 4 Sep 2016

Review of Aksoyoglu et al., "Secondary inorganic aerosols in Europe: Sources and the significant influence of biogenic VOC emissions especially on ammonium nitrate"

In this article, the authors apply CAMx to two intensive monitoring periods, one in the cooler season one in the warmer season. CAMx is applied using its particulate source apportionment technology (PSAT). They double the biogenic emissions inventory to test how that impacts the formation of inorganic aerosol. They find that doubling the biogenics reduces inorganic particulate nitrate. This is tied to the reaction of nitrate radicals with terpenes. They also found that sulfate was mainly of foreign origin.

First, such a paper should have a performance evaluation in the main body of the paper. Simply saying that the chemical components are well captured by the model is not

sufficient. Actual metrics should be provided. They can show this very economically using soccer or bugle plots, along with some traditional performance metrics. The evaluation should consider the available monitors across the modeling domain. In the supplement, they provide only time series pictures, which can be very deceiving depending upon the scales chosen. They should look at all of the work done as part of the AQMEII and follow that lead. Numerical results for the performance on ozone, aerosol nitrate, sulfate and ammonium, and gaseous precursors should be in the text. (I will note, when I look at the time series, it would appear that the model is not performing well, but showing the numerical evaluation would either confirm or negate that view. . . the numerical performance measures should be given either graphically or in a tabular fashion. I think the bugle or soccer plots are best as they can show what is considered reasonable compared to past applications.)

Second, while it is good to also consider periods where intensive measurements are available, it would be good, here, to use annual simulations to limit the bias in interpretation that may be derived from using such short periods. If they were using the detailed measurements to make some process changes in the model, that would be different. Here, the measurements are used in a rather limited fashion. It is also a problem when they only show detailed results for one day (in this case, 14 June). How does this compare to other days. Provide a longer time series or provide a summer and winter average. The limited time period is also of concern when suggesting so much sulfate is coming from the boundaries. This brings up a real concern: is this article meant to support policy decisions or for science (this should be answered in the response to review, not the article). If it is to support policy-making, definitely a longer set of simulations are required. If it is for science, deeper investigation is required (in addition to a longer simulation to show how the period used for more intense investigation represents a typical period). If the period is atypical, that is fine. It just needs to be known.

They used CAMx with PSAT. It should be made clear that PSAT shows where the

species (Nitrate, ammonium, sulfate) originates, but it is not a source impact. If all of the reduced N is removed, most of the oxidized N will also go away. Given the non-linearities in the system, they should also run a series of zero-out simulations, where they zero out the major source categories of interest. These can be compared. This should be computationally quite reasonable.

How do their estimates of nitric acid formation from N2O5 s. OH compare with other historical estimates?

When I go on line, I do not see the "Rest" on their map in the supplement.

Given the huge uncertainties in the NO3-organic and NO2-organic radical and sulfate-BVOC reactions, the finding that doubling the biogenics reduces SIA should be accompanied, quite prominently, this uncertainty. How well does the model reproduce BSOA (biogenic SOA) formation, particularly from terpenes and via the IEPOX pathways? How was this assessed or addressed?

The discussion of NO3 nighttime dynamics lacks context and references, e.g., work done by Seinfeld and co-workers as well as a variety of articles by Platt and coworkers starting, in the early 1980s. They should detail what is new here. This section could also benefit from tracing the HNO3 formed by each reaction. Specifically, while nitric acid is efficiently deposited, the average deposition rate is about 1 cm/s, leading to a lifetime of about a day. It appears more HNO3 is formed during the day, so there is plenty still around at night formed during the day. Keep in mind, the HNO3 formed in the afternoon has little time to deposit. Note how quickly the NO3 raises when the air gets cool enough? The HNo3 is only being formed at a rate of o.o4 ppb/hr, which is likely not fast enough to supply the nitrate shown to be formed. Isn't much of this left over from during the day?

In summary, at present there are a number of items that need to be conducted and/or addressed before the paper should be accepted for publication. First, the model evaluation should be brought forward and discussed, and should include numerical overall

performance measures, potentially shown as soccer and/or bugle charts in the text and a more detailed set of statistics (not just some time series plots) in the supplemental. In particular, the ammonium and nitrate simulations across the domain should be evaluated and considered closely, and the ability of the model to capture BSOA should be brought out. The model should be run to examine how levels respond to removing a few major sources to show how those results compare with the PSAT results. There can be major nonlinearities that are not found when just using PSAT. It would also be advisable to run full year simulations. They need to put their results in context with past studies, e.g., look at the review by Platt and Heinz (1994) and the early work by Seinfeld and co-workers (as well as the recent work, e.g, by Nga et al. 2015). If these items are done in the revision, it would be acceptable for publication.

---

## Referee Comment (RC2) · Anonymous Referee #2 · 20 Sep 2016

The manuscript presents a modeling study to investigate source contributions to PM concentrations in Europe. Using a 3-D model and source apportionment analysis, the authors identified major sources that contribute particulate sulfate, nitrate and ammonium in the modeling domain. They also discussed correlation between biogenic VOC emissions and secondary inorganic PM formation using sensitivity simulations and process analysis. The topic should interest atmospheric modeling community as well as policy-makers. However, there are a couple of issues that need to be addressed before the manuscript should be considered for publication.

Detailed and comprehensive source apportionment analysis is valuable and useful in developing effective air quality management plans. However, it is not clear what scientific contribution this study brings: This study used existing model, modeling database, and analysis tools and methodologies. The authors should clarify/emphasize what their unique and noble contributions are.

The model performance section lacks any quantitative performance evaluation. The authors stated that the model performance has been presented in another paper (Ciarelli et al., 2016), but it appears that Ciarelli et al. mainly evaluated CAMx with a VBS approach while this study used a traditional SOA scheme. In any case, at least some basic statistical performance metrics should be provided. Also, I wonder if any sort of evaluation was done for the boundary conditions from MACC: It seems important considering that BC contributes significantly to sulfate. If manuscript length is a concern, these can be included in the supporting material.

And here are some specific questions: 1. It appears Table 1 doesn't include natural sources (biogenic, wildfires, etc.). They were not considered in the source apportionment analysis? Are their contributions minor?

2. Increased BVOC reduces inorganic nitrate formation, but will increase organic nitrate. What is the overall effect on total PM? Does the model adequately model organic nitrate formation?

3. Figure S10 shows significant nitrate reduction ($\sim$15%) over the ocean while SOA increases are mostly confined inland. There should be no BVOC emissions over the ocean. What is causing nitrate reduction there?
* * *

---

## Author Comment (AC1) · 12 Dec 2016

**Responses to the comments of anonymous referee #1**

Thank you for your comments that helped to improve our manuscript. Please find below your comments in blue, our responses in black and modifications in the revised manuscript in *italic*.

In this article, the authors apply CAMx to two intensive monitoring periods, one in the cooler season one in the warmer season. CAMx is applied using its particulate source apportionment technology (PSAT). They double the biogenic emissions inventory to test how that impacts the formation of inorganic aerosol. They find that doubling the biogenics reduces inorganic particulate nitrate. This is tied to the reaction of nitrate radicals with terpenes. They also found that sulfate was mainly of foreign origin.

First, such a paper should have a performance evaluation in the main body of the paper. Simply saying that the chemical components are well captured by the model is not sufficient. Actual metrics should be provided. They can show this very economically using soccer or bugle plots, along with some traditional performance metrics. The evaluation should consider the available monitors across the modeling domain. In the supplement, they provide only time series pictures, which can be very deceiving depending upon the scales chosen. They should look at all of the work done as part of the AQMEII and follow that lead. Numerical results for the performance on ozone, aerosol nitrate, sulfate and ammonium, and gaseous precursors should be in the text. (I will note, when I look at the time series, it would appear that the model is not performing well, but showing the numerical evaluation would either confirm or negate that view.. the numerical performance measures should be given either graphically or in a tabular fashion. I think the bugle or soccer plots are best as they can show what is considered reasonable compared to past applications.)

Thank you for this comment. The model performance published by Ciarelli et al., (2016) is very similar to the one in this study- since all the model parameters were the same- except for organics due to the difference in organic aerosol model. However we agree with this comment and added the detailed model performance in the revised manuscript (see section 3.1 Model Evaluation) using soccer plots and tables with statistical parameters as suggested.

Second, while it is good to also consider periods where intensive measurements are available, it would be good, here, to use annual simulations to limit the bias in interpretation that may be derived from using such short periods. If they were using the detailed measurements to make some process changes in the model, that would be different. Here, the measurements are used in a rather limited fashion. It is also a problem when they only show detailed results for one day (in this case, 14 June). How does this compare to other days. Provide a longer time series or provide a summer and winter average. The limited time period is also of concern when suggesting so much sulfate is coming from the boundaries. This brings up a real concern: is this article meant to support policy decisions or for science (this should be answered in the response to review, not the article). If it is to support policy-making, definitely a longer set of simulations are required. If it is for science, deeper investigation is required (in addition to a longer simulation to show how the period used for more intense investigation represents a typical period). If the period is atypical, that is fine. It just needs to be known.

This study is not meant for policy decisions for which we have performed annual

simulations in the past (e.g. Aksoyoglu et al., 2014). Here, after showing the significant contribution of various sources to SIA concentrations in Europe, we aim to attract the attention to the role of BVOC emissions in chemical processes leading to formation of SIA, especially ammonium nitrate. For this purpose, short periods with PM component measurements are very useful. Sensitivity tests aim to analyze the changes in the production of OH radical and $HNO_3$ from various chemical reactions in the model and therefore they are shown at one point and one day, as an example. As suggested however, we replaced the figures of sensitivity tests on 14 June with those using monthly averages. These results show the model's response to changes in BVOC emissions and chemical reactions. The periods were chosen based on the availability of AMS measurements of the PM components. These measurements are very valuable for the evaluation of model performance because using only $PM_{2.5}$ measurements for model evaluation might result in a good performance due to compensation of over- and under estimations of PM components. We inserted a table in the revised manuscript showing the model performance evaluation for the PM components.

High sulfate at the eastern boundary is seen mainly during the February-March 2009 period and it affects only the eastern part of the domain (Russia and Ukraine), the rest of Europe doesn't seem to be affected. There are detailed evaluations of MACC reanalysis data (e.g. Inness et al., 2013, Giordano et al., 2015). Evaluations during the AQMEII-2 exercise showed a positive bias for sulfate and suggested that it was because MACC aerosol model does not contain a representation of ammonium nitrate aerosol which represents a large component of the European aerosol loading (Giordano et al., 2015). Therefore the assimilation of satellite AOD will tend to increase the other aerosol components to give the correct AOD overall.

They used CAMx with PSAT. It should be made clear that PSAT shows where the species (Nitrate, ammonium, sulfate) originates, but it is not a source impact. If all of the reduced N is removed, most of the oxidized N will also go away. Given the nonlinearities in the system, they should also run a series of zero-out simulations, where they zero out the major source categories of interest. These can be compared. This should be computationally quite reasonable.

PSAT is a widely used source apportionment model and has already been compared with other methods (e.g. Pirovano et al., 2015, Koo et al., 2009). Studies comparing PSAT which is a reactive tracer method, with sensitivity analysis methods such as brute-force, zero-out and decoupled direct methods pointed out that source apportionment and source sensitivity are not the same thing for nonlinear systems (Yarwood et al., 2007). In PSAT, a single tracer can track primary PM species, whereas secondary PM species require several tracers to track the relationship between gaseous precursors and the resulting PM - for example, in case of nitrogen, PSAT uses tracers such as $NO_X$, $NO_3$ radical, HONO, $N_2O_5$, PAN, PNA, $HNO_3$, organic nitrates, particulate nitrate. Yarwood et al. (2007) compared PSAT and zero-out method results for secondary inorganic aerosols and concluded that PSAT was much more efficient and it was a better approach to source apportionment than zero-out method because it was better able to deal with the nonlinear chemistry. It was recently used to identify source-sector contributions to European fine PM during the Phase 3 of the Air Quality Model Evaluation International Initiative (AQMEII) (Karamchandani et al., 2016).

How do their estimates of nitric acid formation from N2O5 s. OH compare with other

Brown et al. (2004) and Vrekoussis et al. (2006) confirmed the role of $NO_3$ and $N_2O_5$ in producing $HNO_3$ with an efficiency similar to daytime production off the East Coast of the United States and in eastern Mediterranean, respectively. Estimated daytime $HNO_3$ production rate of 2.76 ppb/d and nighttime production (by $N_2O_5$) of 0.21 ppb/d by Minejima (2008) in California are similar to our estimates shown in Fig. 6 in the revised manuscript.

When I go on line, I do not see the "Rest" on their map in the supplement.

"Rest" is shown by the yellow color and it covers all the other countries except those indicated by other colors. Perhaps the tonality of yellow color in the map and the legend were slightly different. We adjusted the legend color and it looks better now.

Given the huge uncertainties in the NO3-organic and NO2-organic radical and sulfate- BVOC reactions, the finding that doubling the biogenics reduces SIA should be accompanied, quite prominently, this uncertainty. How well does the model reproduce BSOA (biogenic SOA) formation, particularly from terpenes and via the IEPOX pathways? How was this assessed or addressed?

We use the Carbon Bond gas-phase mechanism in CAMx which has been developed and updated for EPA atmospheric modeling studies (Yarwood et al., 2005). The CB05 mechanism was evaluated against smog chamber data from the Universities of North Carolina and California at Riverside. Gas-phase reactions of isoprene and its oxidation products as well as reactions of terpenes and SOA precursor reactions are given in the Supplement Tables S1, S2. Isoprene mechanism was revised by the CAMx developers based on Paulot et al. (2009a, b) and Peeters et al. (2009). As seen in Figure S8, BSOA is formed mostly by mono and sesquiterpenes, much less from isoprene. Since SOA is formed mainly from biogenic emissions in Europe, large uncertainties in biogenic VOC emissions are very important (Steinbrecher et al., 2009), might even be more important than the uncertainties in the reactions. As reported by Sartelet et al., (2012), SOA concentrations differ by a factor of 2 using two different biogenic emission inventories. More information was added in the revised introduction.

The discussion of NO3 nighttime dynamics lacks context and references, e.g., work done by Seinfeld and co-workers as well as a variety of articles by Platt and coworkers starting, in the early 1980s. They should detail what is new here.

Nighttime chemistry of $NO_3$ has of course been studied extensively and we added some more references in the Section 3.5 of the revised manuscript as suggested *(Platt et al., 1981; Russell et al., 1986; Platt and Heintz, 1994; Seinfeld and Pandis, 2012)*. Although there is nothing new about the nighttime reactions, this study aims to attract the attention to the consequences of these reactions not only on the formation of organic nitrates and aerosols but also on the inorganic ones. More specifically, we try to show how BVOC reactions play a role on inorganic nitrate formation by consuming nitrate radical. Model studies have so far assumed that other factors such as $NO_x$ and $NH_3$ emissions, deposition and gas-particle partitioning might be responsible for not well capturing ammonium nitrate. We believe that BVOC emissions might also have an important contribution to the model performance of inorganic nitrates. Several studies so far have emphasized the significance of BVOC reactions with nitrate radicals as leading to "anthropogenically influenced biogenic SOA" (Ng et al., 2016). In this study we aim to show another consequence –although

with smaller influence- of such reactions leading to "biogenic influence on anthropogenic ammonium nitrate". Our sensitivity tests with doubled BVOC emissions suggest that terpene reactions reduce the amount of nitrate radical available for nighttime $HNO_3$ formation and consequently, cause a reduction in ammonium nitrate formation.

This section could also benefit from tracing the HNO3 formed by each reaction. Specifically, while nitric acid is efficiently deposited, the average deposition rate is about 1 cm/s, leading to a lifetime of about a day. It appears more HNO3 is formed during the day, so there is plenty still around at night formed during the day. Keep in mind, the HNO3 formed in the afternoon has little time to deposit. Note how quickly the NO3 raises when the air gets cool enough? The HNo3 is only being formed at a rate of o.o4 ppb/hr, which is likely not fast enough to supply the nitrate shown to be formed. Isn't much of this left over from during the day?

In Figure 6 of the revised manuscript, formation rate of $HNO_3$ during the day and night are shown together with the concentration of $PNO_3$ (not the formation rate). Daytime rate of $HNO_3$ formation is faster than nighttime. It is true that some of the $HNO_3$ produced during the day might be left after deposition at night. However, $PNO_3$ concentration shown in Fig. 6, is the result of several processes such as formation from all pathways, transport, deposition, gas-particle partitioning in addition to the nitrate which was already there before.

In summary, at present there are a number of items that need to be conducted and/or addressed before the paper should be accepted for publication. First, the model evaluation should be brought forward and discussed, and should include numerical overall performance measures, potentially shown as soccer and/or bugle charts in the text and a more detailed set of statistics (not just some time series plots) in the supplemental. In particular, the ammonium and nitrate simulations across the domain should be evaluated and considered closely, and the ability of the model to capture BSOA should be brought out. The model should be run to examine how levels respond to removing a few major sources to show how those results compare with the PSAT results. There can be major nonlinearities that are not found when just using PSAT. It would also be advisable to run full year simulations. They need to put their results in context with past studies, e.g., look at the review by Platt and Heinz (1994) and the early work by Seinfeld and co-workers (as well as the recent work, e.g, by Nga et al. 2015). If these items are done in the revision, it would be acceptable for publication.

Thanks for your suggestions. We revised the manuscript with all the points as addressed above individually.

**References**

Aksoyoglu, S., Keller, J., Ciarelli, G., Prévôt, A. S. H., and Baltensperger, U.: A model study on changes of European and Swiss particulate matter, ozone and nitrogen deposition between 1990 and 2020 due to the revised Gothenburg protocol, Atmos. Chem. Phys., 14, 13081-13095, 10.5194/acp-14-13081-2014, 2014.

Brown, S. S., Dibb, J. E., Stark, H., Aldener, M., Vozella, M., Whitlow, S., Williams, E. J., Lerner, B. M., Jakoubek, R., Middlebrook, A. M., DeGouw, J. A., Warneke, C., Goldan, P. D., Kuster, W. C., Angevine, W. M., Sueper, D. T., Quinn, P. K., Bates, T. S., Meagher, J. F., Fehsenfeld, F. C., and Ravishankara, A. R.: Nighttime removal of NOx in the summer marine boundary layer, Geophysical Research Letters, 31, n/an/a, 10.1029/2004GL019412, 2004.

Ciarelli, G., Aksoyoglu, S., Crippa, M., Jimenez, J. L., Nemitz, E., Sellegri, K., Äijälä, M., Carbone, S., Mohr, C., O'Dowd, C., Poulain, L., Baltensperger, U., and Prévôt, A. S. H.: Evaluation of European air quality modelled by CAMx including the volatility basis set scheme, Atmos. Chem. Phys., 16, 10313-10332, 10.5194/acp-16-10313-2016, 2016.

Giordano, L., Brunner, D., Flemming, J., Hogrefe, C., Im, U., Bianconi, R., Badia, A., Balzarini, A., Baró, R., Chemel, C., Curci, G., Forkel, R., Jiménez-Guerrero, P., Hirtl, M., Hodzic, A., Honzak, L., Jorba, O., Knote, C., Kuenen, J. J. P., Makar, P. A., Manders-Groot, A., Neal, L., Pérez, J. L., Pirovano, G., Pouliot, G., San José, R., Savage, N., Schröder, W., Sokhi, R. S., Syrakov, D., Torian, A., Tuccella, P., Werhahn, J., Wolke, R., Yahya, K., Žabkar, R., Zhang, Y., and Galmarini, S.: Assessment of the MACC reanalysis and its influence as chemical boundary conditions for regional air quality modeling in AQMEII-2, Atmospheric Environment, 115, 371-388, http://dx.doi.org/10.1016/j.atmosenv.2015.02.034, 2015.

Inness, A., Baier, F., Benedetti, A., Bouarar, I., Chabrillat, S., Clark, H., Clerbaux, C., Coheur, P., Engelen, R. J., Errera, Q., Flemming, J., George, M., Granier, C., Hadji-Lazaro, J., Huijnen, V., Hurtmans, D., Jones, L., Kaiser, J. W., Kapsomenakis, J., Lefever, K., Leitão, J., Razinger, M., Richter, A., Schultz, M. G., Simmons, A. J., Suttie, M., Stein, O., Thépaut, J. N., Thouret, V., Vrekoussis, M., Zerefos, C., and the, M. t.: The MACC reanalysis: an 8 yr data set of atmospheric composition, Atmos. Chem. Phys., 13, 4073-4109, 10.5194/acp-13-4073-2013, 2013.

Karamchandani, P., Long, Y., Pirovano, G., Balzarini, A., and Yarwood, G.: Source-sector contributions to European ozone and fine PM in 2010 using AQMEII modeling data, Atmos. Chem. Phys. Discuss., 2016, 1-26, 10.5194/acp-2016-973, 2016.

Koo, B., Wilson, G. M., Morris, R. E., Dunker, A. M., and Yarwood, G.: Comparison of Source Apportionment and Sensitivity Analysis in a Particulate Matter Air Quality Model, Environmental Science & Technology, 43, 6669-6675, 10.1021/es9008129, 2009.

Minejima, C.: Nitrogen Oxide Chemistry at Night: Novel Instrumentation Applied to Field Measurements in California, Ph.D. thesis, Department of Chemistry, University of California, Berkeley, CA, USA, 2008.

Ng, N. L., Brown, S. S., Archibald, A. T., Atlas, E., Cohen, R. C., Crowley, J. N., Day, D. A., Donahue, N. M., Fry, J. L., Fuchs, H., Griffin, R. J., Guzman, M. I., Hermann, H., Hodzic, A., Iinuma, Y., Jimenez, J. L., Kiendler-Scharr, A., Lee, B. H., Luecken, D. J., Mao, J., McLaren, R., Mutzel, A., Osthoff, H. D., Ouyang, B., Picquet-Varrault, B., Platt, U., Pye, H. O. T., Rudich, Y., Schwantes, R. H., Shiraiwa, M., Stutz, J., Thornton, J. A., Tilgner, A., Williams, B. J., and Zaveri, R. A.: Nitrate radicals and biogenic volatile organic compounds: oxidation, mechanisms and organic aerosol, Atmos. Chem. Phys. Discuss., 2016, 1-111, 10.5194/acp-2016-734, 2016.

Paulot, F., Crounse, J. D., Kjaergaard, H. G., Kroll, J. H., Seinfeld, J. H., and Wennberg, P. O.: Isoprene photooxidation: new insights into the production of acids and organic nitrates, Atmos. Chem. Phys., 9, 1479-1501, 10.5194/acp-9-1479-2009, 2009a.

Paulot, F., Crounse, J. D., Kjaergaard, H. G., Kürten, A., St. Clair, J. M., Seinfeld, J. H., and Wennberg, P. O.: Unexpected Epoxide Formation in the Gas-Phase

Photooxidation of Isoprene, Science, 325, 730-733, 10.1126/science.1172910, 2009b.

Peeters, J., Nguyen, T. L., and Vereecken, L.: HOx radical regeneration in the oxidation of isoprene, Physical Chemistry Chemical Physics, 11, 5935-5939, 10.1039/B908511D, 2009.

Pirovano, G., Colombi, C., Balzarini, A., Riva, G. M., Gianelle, V., and Lonati, G.: PM2.5 source apportionment in Lombardy (Italy): Comparison of receptor and chemistry-transport modelling results, Atmospheric Environment, 106, 56-70, http://dx.doi.org/10.1016/j.atmosenv.2015.01.073, 2015.

Platt, U., Perner, D., Schröder, J., Kessler, C., and Toennissen, A.: The diurnal variation of NO3, Journal of Geophysical Research: Oceans, 86, 11965-11970, 10.1029/JC086iC12p11965, 1981.

Platt, U., and Heintz, F.: Nitrate Radicals in Tropospheric Chemistry, Israel Journal of Chemistry, 34, 289-300, 10.1002/ijch.199400033, 1994.

Russell, A. G., Cass, G. R., and Seinfeld, J. H.: On some aspects of nighttime atmospheric chemistry, Environmental Science & Technology, 20, 1167-1172, 10.1021/es00153a013, 1986.

Sartelet, K. N., Couvidat, F., Seigneur, C., and Roustan, Y.: Impact of biogenic emissions on air quality over Europe and North America, Atmospheric Environment, 53, 131-141, http://dx.doi.org/10.1016/j.atmosenv.2011.10.046, 2012.

Seinfeld, John H. and Pandis, Spyros N..: Atmospheric Chemistry and Physics: From Air Pollution to Climate Change, 2nd Edition. ISBN: 978-0-471-72018-8, 2012

Steinbrecher, R., Smiatek, G., Köble, R., Seufert, G., Theloke, J., Hauff, K., Ciccioli, P., Vautard, R., and Curci, G.: Intra- and inter-annual variability of VOC emissions from natural and semi-natural vegetation in Europe and neighbouring countries, Atmospheric Environment, 43, 1380-1391, 2009.

Vrekoussis, M., Liakakou, E., Mihalopoulos, N., Kanakidou, M., Crutzen, P. J., and Lelieveld, J.: Formation of HNO3 and NO3− in the anthropogenically-influenced eastern Mediterranean marine boundary layer, Geophysical Research Letters, 33, n/a-n/a, 10.1029/2005GL025069, 2006.

Yarwood, G., Rao, S., Yocke, M., and Whitten, G. Z.: Updates to the Carbon Bond chemical mechanism: CB05 Yocke & Company, Novato, CA 94945RT-04-00675, 2005.

Yarwood, G., Morris, R. E., and Wilson, G. M.: Particulate Matter Source Apportionment Technology (PSAT) in the CAMx Photochemical Grid Model, in: Air Pollution Modeling and Its Application XVII, edited by: Borrego, C., and Norman, A.-L., Springer US, Boston, MA, 478-492, 2007.

---

## Author Comment (AC2) · 12 Dec 2016

**Responses to the comments of anonymous referee #2**

Thank you for your comments on our manuscript. Please find below your comments in blue, our responses in black and modifications in the revised manuscript in *italic*.

The manuscript presents a modeling study to investigate source contributions to PM concentrations in Europe. Using a 3-D model and source apportionment analysis, the authors identified major sources that contribute particulate sulfate, nitrate and ammonium in the modeling domain. They also discussed correlation between biogenic VOC emissions and secondary inorganic PM formation using sensitivity simulations and process analysis. The topic should interest atmospheric modeling community as well as policy-makers. However, there are a couple of issues that need to be addressed before the manuscript should be considered for publication.

Detailed and comprehensive source apportionment analysis is valuable and useful in developing effective air quality management plans. However, it is not clear what scientific contribution this study brings: This study used existing model, modeling database, and analysis tools and methodologies. The authors should clarify/emphasize what their unique and noble contributions are.

The new scientific contribution can be summarized as follows: Several studies so far have emphasized the significance of BVOC reactions with nitrate radicals as leading to "anthropogenically influenced biogenic SOA" (Ng et al., 2016). In this study we aim to show another consequence –although with smaller influence- of such reactions leading to a "biogenic influence on anthropogenic ammonium nitrate". Our sensitivity tests with doubled BVOC emissions suggest that terpene reactions reduce the available nitrate radical for nighttime $HNO_3$ formation and consequently, cause a reduction in ammonium nitrate formation.

In the introduction of the revised manuscript we emphasized the main scientific contribution as shown below.

[revised manuscript text omitted]

The model performance section lacks any quantitative performance evaluation. The authors stated that the model performance has been presented in another paper (Ciarelli et al., 2016), but it appears that Ciarelli et al. mainly evaluated CAMx with a VBS approach while this study used a traditional SOA scheme. In any case, at least some basic statistical performance metrics should be provided. Also, I wonder if any sort of evaluation was done for the boundary conditions from MACC: It seems important considering that BC contributes significantly to sulfate. If manuscript length is a concern, these can be included in the supporting material.

The model performance published by Ciarelli et al., (2016) is very similar to the one in this study- since all the model parameters were the same- except for organics due to the difference in the organic aerosol model. However we agree with this comment and added the detailed model performance in the revised manuscript (see section 3.1 Model Evaluation) using statistical parameters as suggested also by the other referee.

MACC reanalysis data have already been evaluated in detail (e.g. Inness et al., 2013, Giardono et al., 2015). Evaluations during the AQMEII-2 exercise showed a positive bias for sulfate and suggested that it was because the MACC aerosol model does not contain a representation of ammonium nitrate aerosol which represents a large

component of the European aerosol loading (Giardono et al., 2015). Therefore the assimilation of satellite AOD will tend to increase the other aerosol components to give the correct AOD overall. In our study, high sulfate levels at the eastern boundary were mainly during the February-March 2009 period affecting only the eastern part of the domain (Russia and Ukraine).

And here are some specific questions: 1. It appears Table 1 doesn't include natural sources (biogenic, wildfires, etc.). They were not considered in the source apportionment analysis? Are their contributions minor?

Wildfires were not considered in the source apportionment analysis because they would contribute to carbonaceous aerosols, but not much to secondary inorganic aerosols (Gibson et al., 2015). In addition, emission databases for wildfires have usually very low spatial and temporal resolution leading to very high uncertainty in the model predictions. The contribution of other natural sources to fine secondary inorganic aerosols is negligible compared to the anthropogenic sources in Europe.

2. Increased BVOC reduces inorganic nitrate formation, but will increase organic nitrate.

What is the overall effect on total PM? Does the model adequately model organic nitrate formation?

It is true that increased BVOC increases organic nitrates (NTR) through isoprene (ISOP) and isoprene oxidation product (ISPD) reactions with $NO_2$ and $NO_3$ and terpene (TERP) reactions with $NO_3$ (see reactions in Table S1). Organic nitrates may serve as either a NOx reservoir or a NOx sink (Kiendler-Scharr et al., 2016). We inserted also a figure for the production rate of organic nitrates in the revised manuscript (Fig. 6c). Organic nitrate production increased with doubled BVOC emissions. On the other hand, when BVOC reactions with $NO_3$ radical were switched off, organic nitrate production decreased significantly, especially at night due to reduced production from terpene $+NO_3$ reactions.

The overall effect on total $PM_{2.5}$ (up to 5 µg m$^{-3}$, ~65%) is dominated by the increase in SOA (see Figure S9 in the Supplement). A small negative change in $PM_{2.5}$ was predicted around the Benelux area and northern Italy due to the decrease in $PNO_3$ of about 1 -1.4 µg m$^{-3}$ in those regions.

3. Figure S10 shows significant nitrate reduction (_15%) over the ocean while SOA increases are mostly confined inland. There should be no BVOC emissions over the ocean. What is causing nitrate reduction there?

This is just due to a small difference in a small number. As seen in Fig.1 (upper left panel), nitrate concentrations over the ocean are very low and the absolute change (Fig. 4, left panel) is small.

---

## Author Response (AR2)

Responses to the comments of anonymous referee #1

Thank you for your comments. Please find below your comments in blue, our responses in black and modifications in the revised manuscript in *italic*.

Additional comments from reviewer 1:

First, one needs to make sure that one does not get the wrong impression of the importance of daytime vs. nighttime formation of nitric acid leading to aerosol nitrate formation. Doing an eye-ball integration under the curve in figure 5b (now 6b) leads to an estimated HNO3 of about 0.6ppb (or about 1.5 ug m-3). Doing the same for 6c leads to about 0.18 ppb (they might want to do this more precisely). Further, they need to provide further support for their statement "Although most of the HNO3 comes from the daytime reaction of OH with NO2, since the deposition rate of HNO3 is very high, the main pathway leading to the formation of PNO3 is the hydrolysis of N2O5 at night, when the temperature is sufficiently low for partitioning to the particle phase." They could do this by switching of the N2O5 hydrolysis route, or, better yet, have the product of that reaction to be a marked HNO3.

First of all, there was probably a misunderstanding about the figures: The Figs. 6b and 6c in the revised version are not the same as Figs. 5b and 5c in the previous version of the manuscript. The previous Fig. 5b (daytime $HNO_3$ production) and Fig. 5c (nighttime $HNO_3$ production) are Fig. 6a and Fig. 6b, respectively, in the revised manuscript.

We agree completely with the referee that the daytime production of $HNO_3$ is higher than the nighttime production and we have emphasized it in the revised manuscript. The diurnal cycles of $HNO_3$ concentrations usually show a distinct minimum during early morning hours and a maximum in the afternoon with a rapid drop near sunset (Fischer et al., 2006; Aas et al., 2012). The modeled diurnal cycle of $HNO_3$ in this study shows a similar behavior with concentrations in the evening lower than the increase in nitrate concentrations at night (see Fig.1 below). The dry deposition velocity of $HNO_3$ is much higher during the day -when the concentrations are high- than in nighttime (Fischer et al., 2006; Phillips et al., 2006; Zhou et al., 2010). We agree, however, that the deposition is not the only reason of lower concentrations of $HNO_3$ in the evening. We modified the sentence "Although most of the $HNO_3$ comes from the daytime reaction of OH with $NO_2$, since the deposition rate of $HNO_3$ is very high, the main pathway leading to the formation of $PNO_3$ is the hydrolysis of $N_2O_5$ at night, when the temperature is sufficiently low for partitioning to the particle phase." in the Section 3.5 of the revised manuscript as:

*"As seen in Figs. 6a and 6b, the daytime production of $HNO_3$ is higher than the nighttime production. On the other hand, the deposition rate of $HNO_3$ is much higher during the day than in nighttime (Fischer et al., 2006; Phillips et al., 2006; Zhou et al., 2010). $HNO_3$ concentrations usually show a distinct minimum during early morning hours and a maximum in the afternoon with a rapid drop near sunset (Fischer et al., 2006; Aas et al., 2012). The diurnal cycle of the modeled $HNO_3$ in this study shows a similar behavior with concentrations in the evening lower than the increase in nitrate concentrations at night (Fig. S12). Our results suggest that the main pathway leading to the formation of $PNO_3$ is the hydrolysis of $N_2O_5$ at night, when the temperature is*

*sufficiently low for partitioning to the particle phase."*

[Figure]

Figure 1: Diurnal cycle of modeled $HNO_3$ concentrations (ppb) at Payerne (June 2006) (added as Fig. S12 in the revised Supplement).

It also appears Fig. 6d (used to be Fig. 5d) changed markedly. What was the reason for this change? The revised plot decreases the apparent difference.

Fig. 5d in the previous version was only for one day (14 June) and we changed it to an average of the whole month (June) in the revised version (Fig. 6d) following the recommendation of the referees of the previous version. The difference is due to the fact that 14 June was one of the warmest days in June and therefore $PNO_3$ dropped sharply near zero in the early morning hours and remained very low until the sunset when it increased again. As a result of averaging all days, the figure became smoother.

They should still make sure it is clear to the reader that the PSAT result is not the same as the source impact. This is clear in Fig. 7. Given that the nitrate is ammonium nitrate, either the availability of nitrate or ammonium controls formation. In Fig. 7, it shows that most of the NH4 is from SNAP10, while little of the NO3 is from SNAP10 (largest contributor is SNAP7). If one were to remove SNAP10 emissions, much of the PNO3 would be removed. Thus, sentences like "Road transport (SNAP7) was predicted to be the most important source for PNO3 with the largest contribution during the cold season…" should be given context, potentially in the conclusions, as well as further discussion of this when PSAT is originally discussed. For example, the statement "While PSAT quanitifies the source of the the ammonium or nitrate in the PM2.5, this is not the same as the source impact as the system may be limited by another component. For example, removing SNAP10 emissions would not only reduce ammonium, but nitrate as well."

We assume that the referee means not Fig. 7, but Fig. 8 where contributions from various categories to PNO3, PSO4 and PNH4 are shown. One has to

keep in mind that PSAT provides PM attribution to source regions and categories for a given emissions matrix, but does not provide quantitative information as to how PM contributions would change as emissions are altered because chemical interactions are nonlinear. We made a few modifications in the revised text to make this clearer:

In Section 2.2 : *One has to keep in mind that PSAT provides a PM attribution to source regions and categories for a given emissions matrix, but does not provide quantitative information as to how PM contributions would change as emissions are altered because chemical interactions are nonlinear.*

In Conclusions: *One has to keep in mind that these results refer to the emissions matrix used in this study and they might be different if emissions are modified because chemical interactions are nonlinear.*

References

Aas, W., Tsyro, S., Bieber, E., Bergström, R., Ceburnis, D., Ellermann, T., Fagerli, H., Frölich, M., Gehrig, R., Makkonen, U., Nemitz, E., Otjes, R., Perez, N., Perrino, C., Prévôt, A. S. H., Putaud, J. P., Simpson, D., Spindler, G., Vana, M., and Yttri, K. E.: Lessons learnt from the first EMEP intensive measurement periods, Atmos. Chem. Phys., 12, 8073-8094, 10.5194/acp-12-8073-2012, 2012.

Fischer, E., Pszenny, A., Keene, W., Maben, J., Smith, A., Stohl, A., and Talbot, R.: Nitric acid phase partitioning and cycling in the New England coastal atmosphere, Journal of Geophysical Research: Atmospheres, 111, n/a-n/a, 10.1029/2006JD007328, 2006.

Phillips, S., Aneja, V., Kang, D., Arya, S. P.: Modeling and Analysis of the Atmospheric Nitrogen Deposition in North Carolina, *International Journal of Global Environmental Issues*, 6, issue 2/3, 231-252, 2006

Zhou, J., Cui, J., Fan, J.-l., Liang, J.-N., and Wang, T.-J.: Dry deposition velocity of atmospheric nitrogen in a typical red soil agro-ecosystem in Southeastern China, Environmental Monitoring and Assessment, 167, 105-113, 10.1007/s10661-009-1034-2, 2010.

---

## Author Response (AR3)

**Reply to Co-Editor:**

Please find below your comments in blue, our responses in black and modifications in the revised manuscript in *italic*.

Thank you for your response to the further comments by the reviewer. I think you have addressed majority of the comments adequately. However, to strengthen the manuscript, I suggest providing further support for the statement "Our results suggest that the main pathway leading to the formation of PNO3 is the hydrolysis of N2O5 at night, when the temperature is sufficiently low for partitioning to the particle phase", or, re-phrasing the discussion to emphasize the uncertainties. There is N2O5 left over from daytime chemistry. As the reviewer noted, this statement can be evaluated by switching off the hydrolysis of N2O5 in the model. This should be fairly straightforward so I encourage you to consider revising this part of the manuscript to make it stronger/more accurate. (If you prefer not to do this, please include more discussions on the uncertainties).

Thank you for your suggestions. In order to strengthen the manuscript, we performed an additional simulation in which nighttime hydrolysis of $N_2O_5$ was switched off. The results show a significant decrease in $PNO_3$ concentrations at night compared to the standard case at Payerne (Fig. 1 below). The overall effect of hydrolysis reaction during the whole month was predicted to vary between 0.5 and 2.8 µg m$^{-3}$ (about 20 and 46%) in central Europe and UK (Fig. 2).

[Figure]

Figure 1: Change in the diurnal cycle of particulate nitrate concentrations when hydrolysis reaction of $N_2O_5$ was switched off (Payerne, average June 2006).

[Figure]

Figure 2: Change in PNO$_3$ concentrations (upper panel in μg m$^{-3}$, lower panel in %) in June 2006 when hydrolysis reaction of N$_2$O$_5$ was switched off.

On the basis of these additional results, we revised the sentences in lines 273-281 as follows:

*..The diurnal cycle of the modeled HNO$_3$ in this study shows a similar behavior (Fig. S11). Switching off the nighttime hydrolysis of N$_2$O$_5$ in an additional model simulation led to a significant decrease in PNO$_3$ concentrations at night (Fig. 7, upper panel). The overall effect of hydrolysis*

reaction in central Europe in June 2006 was predicted to vary between 20-46% (0.5 and 2.8 $\mu g\ m^{-3}$) as seen in the lower panel of Fig. 7 (see Fig. S12 for the absolute changes). These results suggest that an important part of $HNO_3$ leading to the formation of ammonium nitrate is produced from the hydrolysis of $N_2O_5$ at night. The low temperatures at night then keep the nitrate in the particle phase without evaporating back to $HNO_3$.

[Figure]

*Figure 7: Upper panel: Change in the diurnal cycle of particulate nitrate concentrations when hydrolysis reaction of $N_2O_5$ was switched off (Payerne, average June 2006) Lower panel: Change in $PNO_3$ concentrations (in %) in whole model domain when hydrolysis reaction of $N_2O_5$ was switched off (June 2006).*

[Figure]

*Figure S12: Change in PNO₃ concentrations (in μg m⁻³) in whole model domain when hydrolysis reaction of N₂O₅ was switched off (June 2006).*

On a different note, I suggest including Xu et al. (PNAS, 2015), Xu et al. (ACP, 2015), and Kiendler-Scharr et al. (GRL, 2016) in the citation for the sentence "Reactions of isoprene lead to the formation of SOA mainly during the daytime while nighttime oxidation of monoterpenes by the nitrate radical is responsible for organic nitrate formation" in the introduction, which are all relevant to the discussions here.

We added the suggested references in line 65.

[revised manuscript text omitted]

805    **Figure S12: Change in PNO$_3$ concentrations (in µg m$^{-3}$) in June 2006 when hydrolysis reaction of N$_2$O$_5$ was switched off.**

[Figure]

**Figure S13: Changes in diurnal variation of reacted OH radical (June 2006, Payerne). Black: with standard BVOC emissions, red: with doubled BVOC emissions.**

810